# Detecting and Approximating Redundant Computational Blocks in Neural Networks

## Abstract

Deep neural networks often learn similar internal representations, both across different models and within their own layers. While inter-network similarities have enabled techniques such as model stitching and merging, intra-network similarities present new opportunities for designing more efficient architectures. In this paper, we investigate the emergence of these internal similarities across different layers in diverse neural architectures, showing that similarity patterns emerge independently of the datataset used. We introduce a simple metric, Block Redundancy, to detect redundant blocks, providing a foundation for future architectural optimization methods. Building on this, we propose Redundant Blocks Approximation (RBA), a general framework that identifies and approximates one or more redundant computational blocks using simpler transformations. We show that the transformation $\mathcal{T}$ between two representations can be efficiently computed in closed-form, and it is enough to replace the redundant blocks from the network. RBA reduces model parameters and time complexity while maintaining good performance. We validate our method on classification tasks in the vision domain, using a variety of pretrained foundational models and datasets.

## 1 Introduction

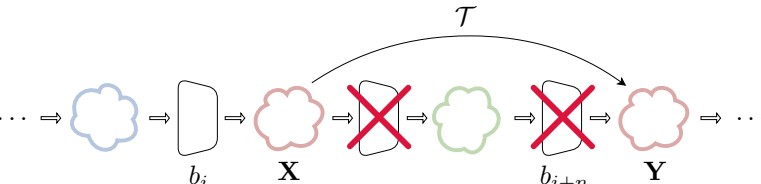

Figure 1: **Framework Description**. Given two latent spaces $\mathbf{X}$ and $\mathbf{Y}$ representing respectively the output of blocks $b_i$ and $b_{i+n}$ for a subset of $n$ data points from the training set, we approximate a transformation matrix $\mathcal{T}$ such that: $\mathbf{Y} \approx \mathbf{Y}' = \mathcal{T}(\mathbf{X})$ to recover a representation $\mathbf{Y}' \approx \mathbf{Y}$.

As Neural Networks (NNs) grow in size and complexity, their demand for computational resources has become a significant bottleneck. Despite the impressive performance of large models, they often come with substantial trade-offs, such as slower inference times and increased memory and power consumption. This has led to a growing interest in methods that can reduce model complexity without sacrificing performance. However, most approaches to mitigating these challenges either require additional training or complex fine-tuning, or they result in a non-trivial loss in performance. However, recent research showed that there exists internal representation similarities within and between NNs. Thus, many layers or components within these networks may perform similar functions or yield highly correlated outputs, suggesting the potential for simplifying these networks. Understanding and leveraging these internal similarities can open up new opportunities for reducing model size, enhancing inference speed, and improving computational efficiency.

In this paper, we address two key research questions: (i) how to identify redundant blocks, and (ii) how to effectively approximate these blocks while preserving the final representations and the network's overall functionality. To address the first question, we introduce a straightforward metric, the Block Redundancy (BR) score, which helps identifying components that do not contribute significantly to the network's final representation. By carefully selecting which blocks to approximate, we can ensure

minimal impact on the network's final output. For the second question, we propose the Redundant Blocks Approximation (RBA), a novel method that leverages internal representation similarities to approximate redundant computational blocks using lightweight transformations, such as linear mappings. Once the blocks that have minimal impact on model functionality are identified, instead of using these redundant blocks in each forward pass (e.g., transformer blocks containing attention and normalization operations), RBA completely replaces them with a simpler transformation. Thanks to this approximation, RBA reduces model parameters and accelerates inference while maintaining the integrity of the final representation produced by the original model.

Our main contributions are as follows:

- We provide a comprehensive analysis of internal representation similarities across various pretrained foundation models, revealing consistent patterns between blocks within each architecture, independent of the dataset (Figures 2 and 8 to 9).
- We show that a simple metric such as the MSE is enough for assessing the redundancy of individual blocks within a NN (Figure 3).
- We introduce RBA, a general framework for identifying and approximating redundant computational blocks in NNs using simpler transformations (e.g., linear), reducing model parameters and complexity with minimal to no impact on the produced representations (Figure 1).
- We validate our method on vision-based classification tasks using diverse pretrained models and datasets, demonstrating its applicability and effectiveness across different architectures and datasets (Tables 1, 7 and 8).

## 2 RELATED WORK

**Measuring Similarities.** A range of metrics have been introduced to assess the similarity between latent spaces generated by different NNs Klabunde et al. (2023); Ballester et al. (2023). One established approach is Canonical Correlation Analysis (CCA) (Hotelling, 1992), known for its invariance to linear transformations. Variants of CCA, such Singular Value Decomposition (SVD) and Singular Value CCA (SVCCA) (Raghu et al., 2017), aim to enhance robustness, while techniques like Projection Weighted CCA (PWCCA) (Morcos et al., 2018) mitigate sensitivity to small perturbations. Another widely used metric, Centered Kernel Alignment (CKA) (Kornblith et al., 2019), captures the similarity between latent spaces while ignoring orthogonal transformations. However, recent work (Davari et al., 2022) highlights that this metric can be sensitive to shifts in the latent space. Additionally, Barannikov et al. (2021) proposes a method to compare two data representations by measuring the multi-scale topological dissimilarity, while Fumero et al. (2024) leverages the principles of spectral geometry to model and analyze the relationships between distinct latent spaces.

**Leveraging Similarities.** Analyzing the similarities between internal representations, both within and across NNs, has received significant attention in recent research. Valeriani et al. (2024) examines the intrinsic dimensions and neighbor compositions of representations in various transformer models. Similarly, Kvinge et al. (2022) explores how models process different variations of data points across layers, while Nguyen et al. (2020) investigates how changes in network depth and width impact hidden representations, revealing characteristic block structures. Finally, Crisostomi et al. (2023) investigates under what assumptions two latent spaces be merged into one. All these insights have been applied across various contexts. Moschella et al. (2023) constructs a unified space shared by different NNs, enabling zero-shot stitching of independently trained models across different modalities Norelli et al. (2023), even without explicit assumptions on the transformation class that connects the latent manifold embeddings Cannistraci et al. (2024) or with partial correspondence within the latent spaces Cannistraci et al. (2023). While Ricciardi et al. (2023) proves the feasibility of zero-shot stitching between encoders and policies trained on different environmental variations. Other works Lähner & Moeller (2024); Maiorca et al. (2024) demonstrate that representations learned by distinct NNs can be aligned using simple transformations. Finally, Tang et al. (2023) leverages similarities in unified visual-language models to dynamically skip layers in both encoders and decoders.

**Architectural Efficiency.** While large-scale models with billions or even trillions of parameters continue to achieve state-of-the-art performance, their growth comes with trade-offs, including slower inference times and significantly higher computational costs. To address these issues, various

techniques have been developed, such as early exiting and model pruning. Early exit strategies, which introduce intermediate output layers at different stages of the network, have been shown to improve efficiency and reduce inference time (Xin et al., 2020; Zhou et al., 2020; Yu et al., 2022). However, this approach requires the additional training of intermediate classifiers to enable exits at predefined layers. On the other hand, model pruning reduces the computational load of Deep Neural Network (DNN) by either removing individual weights based on certain criteria (Ma et al., 2023; Liao et al., 2023) or eliminating or compressing larger structural components such as channels or attention heads (Zhang & He, 2020; Sajjad et al., 2023; Venkataramanan et al., 2024; Zhang et al., 2024; Bai et al., 2023). Although effective, this approach usually requires first training the full model in its dense form, followed by multiple iterations of pruning and retraining or training the pruned model from scratch.

Instead of removing layers or components, we focus on identifying redundant computational blocks within the network and replacing them with lightweight transformations. Unlike other approaches, RBA is an *architecture-agnostic* method to reduce model complexity and computational overhead *without the need for additional training or fine-tuning* while still maintaining competitive performance.

## 3 REDUNDANT BLOCKS APPROXIMATION

The core principle of our approach, RBA, is to detect similar representations within NNs, identifying redundant blocks, and approximate them with simpler transformations instead of executing the entire DNN. A visual overview is provided in Figure 1.

In this section, we first show how to identify redundant blocks, and how to effectively approximate their representations while preserving the network's overall functionality.

**Identifying Redundant Representations.** We hypothesize that certain foundation model architectures, such as Vision Transformers (ViTs), may contain redundant blocks that produce similar representations. This redundancy may stem from overparameterization or task-specific characteristics. In this context, a "block" refers to a self-contained unit in the model that typically contains several layers, such as self-attention, normalization, or feed-forward layers, but functions as a cohesive unit.

To quantify redundancy, we introduce a simple metric called Block Redundancy (BR), which measures the degree of change in internal representations between blocks. This helps to identify essential blocks versus those that contribute minimally to the overall model.

Let $B$ represent the total number of blocks in the model, and let $\mathbf{h}^{(b)}$ denote the internal representation (i.e., the output) of block $b$, where $b \in 1, 2, \ldots, B$. For a given subset of the training data $\mathcal{D}_{\text{sub}}$, we compute the representations $\mathbf{h}^{(b)}(x)$ for each input $x \in \mathcal{D}_{\text{sub}}$. The BR for block $b$ is defined as the negative Mean Squared Error (MSE) between the output representations of blocks $b$ and $b - 1$:

$$\text{BR}(b) = -\frac{1}{|\mathcal{D}_{\text{sub}}|} \sum_{x \in \mathcal{D}_{\text{sub}}} \left\| \mathbf{h}^{(b)}(x) - \mathbf{h}^{(b-1)}(x) \right\|_2^2 \tag{1}$$

A higher $\text{BR}(b)$ indicates a minimal change between the outputs of block $b$ and the preceding block $b - 1$, suggesting a potential redundancy in block $b$. Conversely, a lower $\text{BR}(b)$ implies that block $b$ plays a significant role in transforming the internal representations.

By systematically evaluating the BR for each block, we can identify redundant components that can be simplified, enabling a reduction in the NN's complexity without compromising the original final representation or its performance.

**Approximating Redundant Blocks.** After identifying redundant representations using BR, the next step is to approximate their outputs through more computationally efficient transformations, rather than directly removing the blocks. While this approach applies to consecutive blocks such as $b_i$ and $b_{i+1}$, it generalizes naturally to non-consecutive blocks as well. Specifically, for any block $b_i$ and block $b_{i+n}$ (where $n \geq 1$), our method enables the approximation of the output of block $b_{i+n}$ from the output of block $b_i$, provided they exhibit low BR scores. This allows us to skip the computation of blocks $b_{i+1}, b_{i+2}, \ldots, b_{i+n}$, effectively reducing the overall computation.

Let $\mathbf{X} \in \mathbb{R}^{n \times d_1}$ represent the output of block $b_i$ for a subset of $n$ data points from the training set, where $d_1$ is the dimensionality of the latent space. Similarly, let $\mathbf{Y} \in \mathbb{R}^{n \times d_2}$ represent the output of

block $b_{i+n}$ for the same subset of data points, with $d_2$ being the dimensionality of the latent space at block $b_{i+n}$. Our objective is to find a function $\mathcal{T} : \mathbb{R}^{d_1} \to \mathbb{R}^{d_2}$ such that:

$$\mathbf{Y} \approx \mathcal{T}(\mathbf{X})$$

In this work, we consider $\mathcal{T}$ to be a linear transformation ($\mathbf{T}$) that can be estimated by minimizing the squared error between the transformed output of block $b_i$ and the actual output of block $b_{i+n}$, which can be solved using least squares:

$$\mathbf{T} = \arg\min_{\mathcal{T}} \|\mathbf{Y} - \mathcal{T}(\mathbf{X})\|_2^2$$

This optimization problem allows for a closed-form solution that efficiently computes the optimal transformation $\mathbf{T}$. The solution bypasses the computation of any redundant blocks between $b_i$ and $b_{i+n}$, replacing them with $\mathbf{T}$. This approximation results in a significant reduction in computational complexity, as one or more full transformer block consisting of multi-head self-attention and feed-forward layers can be replaced by a low-cost linear transformation.

To sum up, the overall pipeline of our approach comprises two main stages:

1. **Redundancy Identification:** We apply the BR metric to identify redundant blocks across the model based on their contribution to the transformation of internal representations.

2. **Block Approximation:** For blocks deemed redundant, we compute an efficient linear approximation, using the transformation matrix $\mathbf{T}$ to bypass these blocks.

This process reduces model parameters and computational complexity with minimal impact on the resulting representations, as shown in Figures 3, 4 and 10 to 13. Additionally, it is possible train any downstream linear classifier on top of the simplified model for the desired task, retaining the original architecture's overall structure while significantly decreasing the number of parameters and computation costs, as shown in Tables 1, 2 and 7 to 9.

## 4 EXPERIMENTS

In this section, we analyze the representation of foundation pre-trained models and we show quantitative experiments to evaluate the effectiveness of our proposed framework. We begin by empirically motivating our study in Section 4.1, where we analyze the similarity between different blocks of pretrained foundation models for image classification. Then in Section 4.2, we assess the impact of approximating blocks on latent representations and explore the correlation between layer approximations and high BR. Finally in Section 4.3, we conduct quantitative experiments on the image classification task to further evaluate the performance of our framework across various models and datasets, demonstrating its general applicability and effectiveness.

### 4.1 BLOCK SIMILARITIES

**Experimental Setting.** In this section, we analyze the latent spaces generated by pretrained foundational models in the vision domain. Our analysis focuses on five distinct transformer-based models: `ViT-S`, `ViT-B`, `DiNO-S`, `DiNO-B`, and `DEiT-S`. We evaluate their similarities using four well-known datasets: `CIFAR-10`, `CIFAR-100` (Krizhevsky et al., 2009), `MNIST` (Deng, 2012), and `F-MNIST` (Xiao et al., 2017). Since these models classify input based on the representation of the [CLS] token, the analysis is conducted using the [CLS] token from each block, rather than the full representation. This ensures that the analysis remains aligned with the key components of the model's final predictions. This flexibility enables the method to adapt to different model architectures and tasks, where tokens other than the [CLS] may hold more relevant information. Model and dataset details can be found in Table 5 and Table 6, respectively.

**Results and Analysis.** Figure 2 presents the cosine similarity matrices between blocks of the `ViT-B` and `DiNO-S` models on `MNIST` and `CIFAR-100`. These matrices illustrate the internal block-by-block similarities within each architecture. Our results reveal that while the patterns of similarity vary across architectures, they remain consistent across different datasets. This suggests that the similarity structure between computational blocks is predominantly influenced by the model architecture itself, rather than the specific dataset used. This finding aligns with observations from Nguyen et al. (2020),

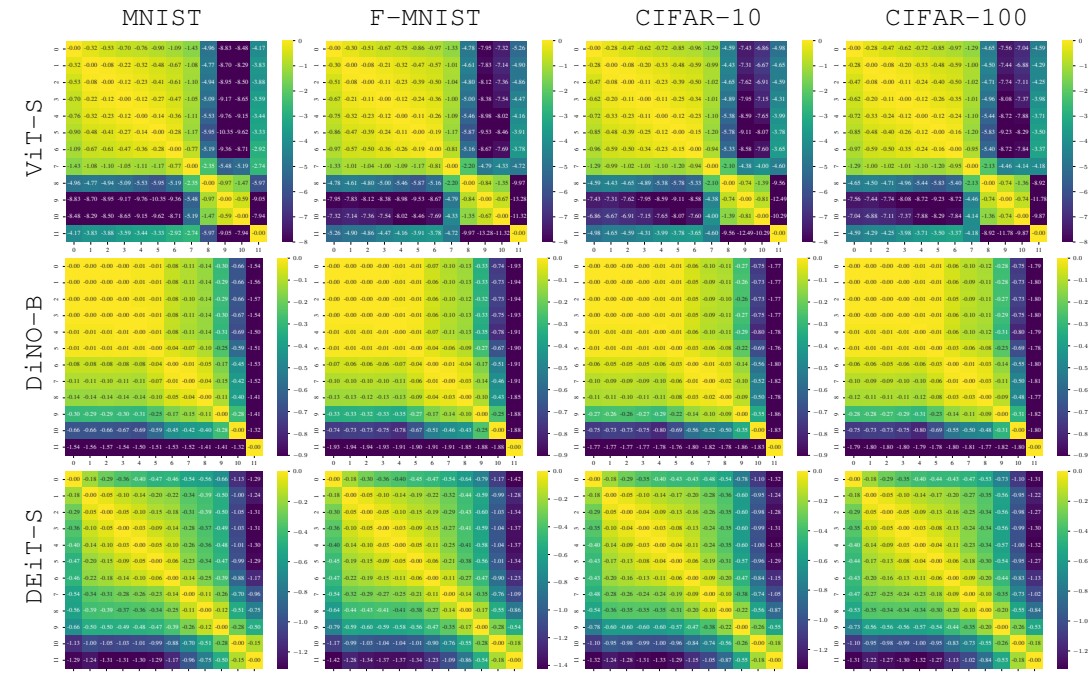

Figure 2: **Representation Redundancies.** BR matrices illustrating the internal block-by-block redundancies in `ViT-S`, `DiNO-B`, and `DEiT-S` models across four datasets: `MNIST`, `F-MNIST`, `CIFAR-10`, and `CIFAR-100`. Each heatmap quantifies the BR metric between internal representations of different blocks using the Classify token ([CLS]) token, providing insights into redundancy in foundation pretrained models. The matrices reveal that the similarity structure between computational blocks is predominantly influenced by the model architecture itself, rather than the specific dataset. Please refer to Figures 8 and 9 for additional results using other metrics and models.

where wide and deep trained from scratch models tend to exhibit a distinctive "block structure" in their representations, linked to model overparameterization. Our results extend this observation by showing that block structures also emerge in pretrained foundation models, with their presence primarily dependent on the architecture. Please refer to Figures 8 and 9 for additional results.

**Takeaway.** The representation patterns generated by pretrained models are primarily determined by the architecture, and remain consistent across different datasets.

## 4.2 REDUDANT BLOCK APPROXIMATION

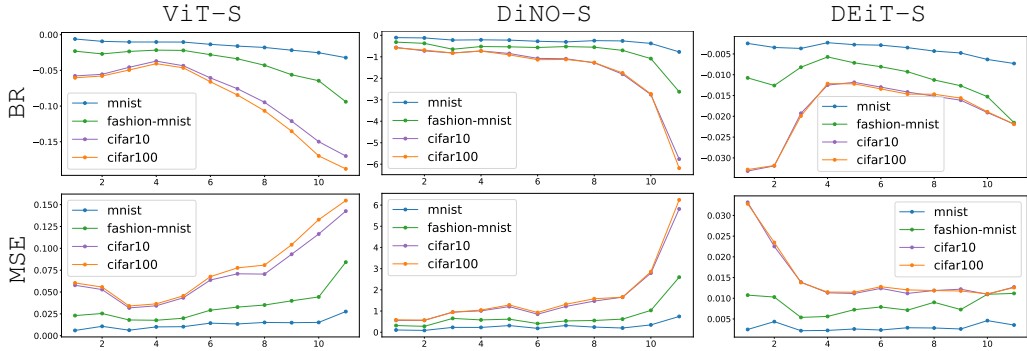

Figure 3: **Block Redundancy vs. Representation Similarity.** This figure illustrates the correlation between the BR metric when approximating the $i^{th}$ block and the MSE between the last layer representations of the original encoder and the approximated encoder. Each column corresponds to a different model (`ViT-S`, `DiNO-S`, `DEiT-S`), while the various curves represent different datasets.

**Experimental Setting.** In Section 4.1, we empirically demonstrate that different blocks in pretrained models exhibit similarities. To further investigate this, introduce the Block Redundancy metric. As illustrated in Equation (1), this metric measures the level of redundancy of a block: a high score indicates minimal change between two blocks output, suggesting that the second block may be redundant. Conversely, a low score implies that the second block contributes significantly to the final prediction. After identifying redundant blocks, we restructure the models accordingly to reduce their complexity and parameter count. These redundant blocks are approximated using a shared linear transformation applied across all tokens, based on a subset of 3,000 training samples. We compute BR scores for each block across different datasets and pretrained encoders: `ViT-S`, `DiNO-S`, `DEiT-S`, utilizing `MNIST`, `F-MNIST`, `CIFAR-10`, and `CIFAR-100`. Additionally, we compute the MSE between the representations of the last layer in the original model and the RBA model when skipping the $i^{th}$ block. We also visualize the Principal Component Analysis (PCA) projections of these representations when specific blocks are approximated to assess the impact on representation fidelity.

**Quantitative Analysis.** As illustrated in Figure 3, in most cases, the BR decreases as the block depth increases. This suggests that approximating the final blocks would lead to significant changes in the final representations, indicating their critical role in maintaining similar final representations. However, in the case of `DEiT-S`, the trend is reversed. Here, the BR is higher in the central blocks and lower in the initial ones. This is confirmed by the dissimilarity between the last-layer representations, which increases when the earlier blocks are removed in `DEiT-S`, whereas the opposite is observed in other models. These findings reinforce the intuition behind the BR metric, demonstrating a correlation between BR and the final representation similarity when approximating blocks. In some instances, such as with the `MNIST` dataset, the BR scores remain relatively consistent across blocks, indicating that the representations are largely similar one to another. However, for more complex datasets like `CIFAR-100`, the representations in the final or in the first blocks become increasingly dissimilar, making it advantageous to approximate intermediate blocks. This suggests that the BR metric is influenced not only by the architecture but also by the complexity of the dataset, allowing for targeted approximations that reduce model parameters and complexity without significantly compromising performance.

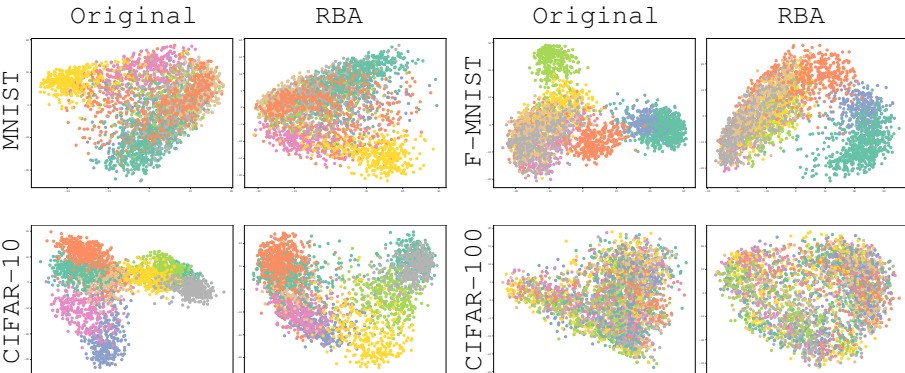

Figure 4: **Last Block Approximation.** PCA visualization of the final layer representations for both the original model and the model with its last block approximated from the preceding one. The representations are generated using the `DiNO-S` model across four datasets. The plots highlight that in this model, the last layer representations are crucial, making it more effective to approximate earlier blocks instead. Note that for `CIFAR-100` (bottom right), only the overall structure of the space can be observed, as the 100 classes make it challenging to distinguish labels based on color. For further results approximating other blocks and using other encoders, refer to Figures 10 to 12.

**Qualitative Analysis.** To further investigate the relationship between BR and representation (dis)similarity, Figure 4 and Figure 5 show the PCA projection of the final block's representations in both the original and approximated models, with a focus on approximating the $11^{th}$ block. These plots visualize the representations generated using the `DiNO-S` and `DEiT-S` pretrained encoders across the `MNIST`, `F-MNIST`, `CIFAR-10`, and `CIFAR-100` datasets. For `CIFAR-10`, having 100 classes, only the overall structure of the representation space is visible, making it difficult to distinguish individual labels by color. In Figure 4, approximating the final block results in noticeable deviations from the original representations, while in Figure 5, the approximated representation

remains similar to the original one. This observation aligns with the results from Figure 3, where approximating the appropriate block can lead to significant changes in representations. Finally, in Figure 7, we present an ablation study on various similarity metrics, analyzing their correlation with downstream accuracy. The results demonstrate that the BR metric is particularly effective in identifying the optimal blocks for approximation. For additional visualizations, please refer to Figures 10 to 13.

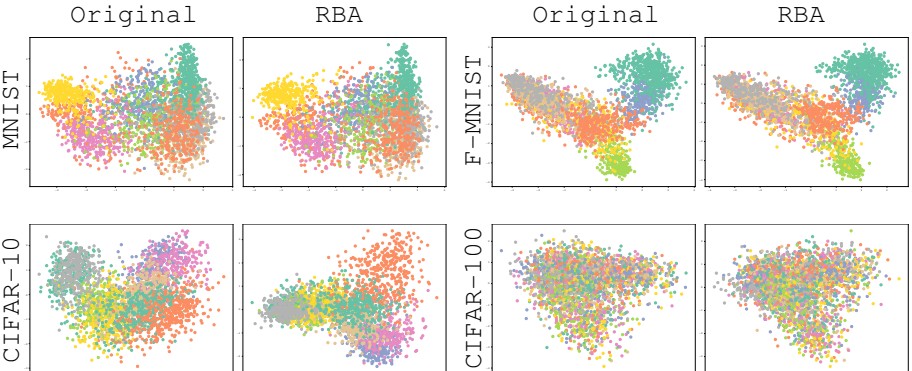

Figure 5: **Last Block Approximation.** PCA visualization of the final layer representations for both the original model and the model with its last block approximated by the preceding one. The representations are generated using the `DEiT-S` model across four datasets. The plots highlight that in this model, the representations in the last layer are redundant and can be effectively approximated, offering potential performance improvements while reducing model complexity and parameter count. Note that for `CIFAR-100` (bottom right), only the overall structure of the space can be observed, as the 100 classes make it challenging to distinguish labels based on color. For further results approximating other blocks and using other encoders, refer to Figures 10 to 12.

**Takeaway.** Approximating redundant blocks effectively reduces model parameters and complexity without significantly compromising representation fidelity.

### 4.3 DOWNSTREAM TASK: CLASSIFICATION

**Experimental Setting.** We finally conduct image classification using the same datasets and pretrained models described in previous sections, with all models remaining pretrained and frozen. After identifying redundant blocks, the models are restructured accordingly. Approximations between blocks are computed using a shared linear transformation across all tokens, based on a subset of 3,000 training samples. Subsequently, a single linear layer is trained for classification using the Adam optimizer with a learning rate of $0.001$ over 5 epochs, three seeds, and a batch size of 256.

| Approx. | Num. Params | ImageNet1k |
|---|---|---|
| $1 \rightarrow 5$ | 15.31M | $43.68 \pm 0.36$ |
| $2 \rightarrow 5$ | 16.94M | $60.41 \pm 0.06$ |
| $7 \rightarrow 10$ | 16.94M | $33.77 \pm 0.44$ |
| $1 \rightarrow 3$ | 18.56M | $65.31 \pm 0.14$ |
| $3 \rightarrow 5$ | 18.56M | $68.16 \pm 0.16$ |
| $2 \rightarrow 4$ | 18.56M | $67.81 \pm 0.15$ |
| $8 \rightarrow 10$ | 18.56M | $46.75 \pm 0.21$ |
| $9 \rightarrow 11$ | 18.56M | $46.17 \pm 0.25$ |
| $2 \rightarrow 3$ | 20.19M | $\mathbf{71.74 \pm 0.29}$ |
| $3 \rightarrow 4$ | 20.19M | $71.70 \pm 0.28$ |
| $4 \rightarrow 5$ | 20.19M | $71.49 \pm 0.23$ |
| $9 \rightarrow 10$ | 20.19M | $61.11 \pm 0.15$ |
| - | 21.82M | $\underline{73.98 \pm 0.19}$ |

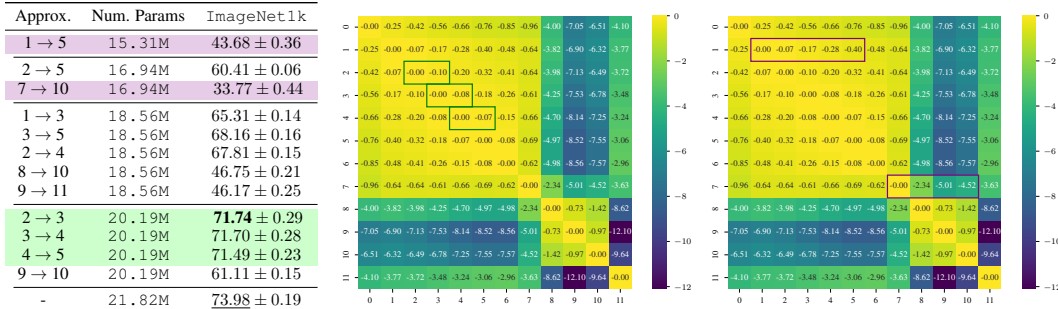

Figure 6: **BR and Accuracy Approximation Correlation.** (*Left*) Accuracy performance of the `ViT-S` encoder with various approximation strategies on `ImageNet1k`. (*Right*) The block-by-block BR matrix. Results highlighted in *green* demonstrate that approximating blocks with high BR values maintains comparable accuracy while reducing parameter count and speeding up computations. Comparatively, results in *purple* show that approximating four high-BR blocks yields better accuracy than approximating three low-BR blocks, which exhibit lower redundancy.

Table 1: **Image Classification Performance Across Architectures and Seeds.** Classification accuracy scores for `ViT-S`, `DiNO-S` and `DEiT-S` using `MNIST`, `CIFAR-10` and `CIFAR-100C`, and 3 random seeds. `CIFAR-100C` refers to `CIFAR-100` with the `coarse` setting (20 labels). The "Approx" column $b_i \rightarrow b_i + n$ specifies the blocks used for approximation, where the first value represents the block whose output is used to approximate the second block's output. The "Num. Blocks" column indicates the total number of remaining blocks after the approximation, and the "Num. Params" column shows the number of model parameters. The proposed method preserves performance while reducing the number of parameters. Please refer to Table 7 for the results on all the models and datasets, as well as Table 8.

| Encoder | Approx. | Num. Blocks | Num. Params | Accuracy ↑ | | | |
|---|---|---|---|---|---|---|---|
| | | | | MNIST | CIFAR-10 | CIFAR-100C | ImageNet1k |
| ViT-S | $1 \rightarrow 5$ | 8 | 15.31M | $92.11 \pm 0.20$ | $84.93 \pm 0.62$ | $68.47 \pm 0.30$ | $43.68 \pm 0.36$ |
| | $2 \rightarrow 5$ | 9 | 16.94M | $94.67 \pm 0.12$ | $90.97 \pm 0.30$ | $78.07 \pm 0.38$ | $60.41 \pm 0.06$ |
| | $7 \rightarrow 10$ | 9 | 16.94M | $94.91 \pm 0.30$ | $85.81 \pm 1.03$ | $71.10 \pm 0.51$ | $33.77 \pm 0.44$ |
| | $1 \rightarrow 3$ | 10 | 18.56M | $95.67 \pm 0.19$ | $92.09 \pm 0.30$ | $79.68 \pm 0.20$ | $65.31 \pm 0.14$ |
| | $2 \rightarrow 4$ | 10 | 18.56M | $95.37 \pm 0.08$ | $93.03 \pm 0.10$ | $81.74 \pm 0.28$ | $67.81 \pm 0.15$ |
| | $9 \rightarrow 11$ | 10 | 18.56M | $94.77 \pm 0.10$ | $89.16 \pm 1.10$ | $75.30 \pm 0.44$ | $46.17 \pm 0.25$ |
| | $2 \rightarrow 3$ | 11 | 20.19M | $\mathbf{95.76} \pm 0.08$ | $94.87 \pm 0.20$ | $85.96 \pm 0.05$ | $\mathbf{71.74} \pm 0.29$ |
| | $3 \rightarrow 4$ | 11 | 20.19M | $95.70 \pm 0.11$ | $95.10 \pm 0.23$ | $86.00 \pm 0.12$ | $71.70 \pm 0.28$ |
| | $4 \rightarrow 5$ | 11 | 20.19M | $95.67 \pm 0.17$ | $\mathbf{95.43} \pm 0.25$ | $\mathbf{86.24} \pm 0.21$ | $71.49 \pm 0.23$ |
| | $9 \rightarrow 10$ | 11 | 20.19M | $95.75 \pm 0.44$ | $94.23 \pm 0.12$ | $82.69 \pm 0.49$ | $61.11 \pm 0.15$ |
| | - | 12 | 21.82M | $\underline{95.95} \pm 0.40$ | $\underline{95.87} \pm 0.08$ | $\underline{87.60} \pm 0.15$ | $\underline{73.98} \pm 0.19$ |
| DiNO-S | $1 \rightarrow 5$ | 8 | 15.55M | $95.32 \pm 1.09$ | $79.37 \pm 1.34$ | $60.72 \pm 0.49$ | $19.45 \pm 0.64$ |
| | $2 \rightarrow 5$ | 9 | 17.18M | $96.04 \pm 0.67$ | $85.58 \pm 0.54$ | $67.89 \pm 0.57$ | $41.39 \pm 0.17$ |
| | $7 \rightarrow 10$ | 9 | 17.18M | $96.93 \pm 0.45$ | $91.24 \pm 0.13$ | $78.14 \pm 0.14$ | $45.94 \pm 0.40$ |
| | $1 \rightarrow 3$ | 10 | 18.80M | $96.74 \pm 0.96$ | $91.82 \pm 0.17$ | $78.81 \pm 0.35$ | $57.38 \pm 0.13$ |
| | $2 \rightarrow 4$ | 10 | 18.80M | $96.54 \pm 0.55$ | $91.03 \pm 0.75$ | $76.57 \pm 0.25$ | $60.26 \pm 0.26$ |
| | $9 \rightarrow 11$ | 10 | 18.80M | $92.46 \pm 1.63$ | $85.65 \pm 0.68$ | $72.44 \pm 1.19$ | $34.50 \pm 0.10$ |
| | $2 \rightarrow 3$ | 11 | 20.43M | $96.99 \pm 0.70$ | $94.67 \pm 0.20$ | $83.92 \pm 0.49$ | $65.42 \pm 0.25$ |
| | $3 \rightarrow 4$ | 11 | 20.43M | $97.22 \pm 0.50$ | $\mathbf{94.72} \pm 0.24$ | $83.37 \pm 0.37$ | $\mathbf{65.60} \pm 0.39$ |
| | $4 \rightarrow 5$ | 11 | 20.43M | $\mathbf{97.33} \pm 0.47$ | $94.64 \pm 0.10$ | $82.81 \pm 0.62$ | $64.58 \pm 0.30$ |
| | $9 \rightarrow 10$ | 11 | 20.43M | $96.99 \pm 0.97$ | $93.52 \pm 0.48$ | $\mathbf{84.09} \pm 0.52$ | $59.19 \pm 0.10$ |
| | - | 12 | 22.06M | $\underline{96.85} \pm 1.04$ | $\underline{96.06} \pm 0.32$ | $\underline{87.62} \pm 0.24$ | $\underline{67.74} \pm 0.23$ |
| DEiT-S | $1 \rightarrow 5$ | 8 | 15.31M | $93.27 \pm 0.37$ | $78.20 \pm 0.21$ | $59.82 \pm 0.16$ | $43.37 \pm 0.18$ |
| | $2 \rightarrow 5$ | 9 | 16.94M | $94.99 \pm 0.18$ | $85.27 \pm 0.11$ | $69.95 \pm 0.15$ | $61.67 \pm 0.16$ |
| | $7 \rightarrow 10$ | 9 | 16.94M | $95.81 \pm 0.23$ | $89.20 \pm 0.34$ | $75.96 \pm 0.20$ | $57.10 \pm 0.22$ |
| | $1 \rightarrow 3$ | 10 | 18.56M | $95.35 \pm 0.21$ | $85.59 \pm 0.23$ | $70.61 \pm 0.42$ | $66.05 \pm 0.26$ |
| | $2 \rightarrow 4$ | 10 | 18.56M | $95.68 \pm 0.11$ | $88.76 \pm 0.08$ | $75.83 \pm 0.38$ | $69.96 \pm 0.12$ |
| | $9 \rightarrow 11$ | 10 | 18.56M | $95.64 \pm 0.13$ | $\mathbf{91.09} \pm 0.21$ | $79.30 \pm 0.58$ | $69.63 \pm 0.24$ |
| | $2 \rightarrow 3$ | 11 | 20.19M | $95.99 \pm 0.19$ | $90.13 \pm 0.23$ | $78.11 \pm 0.23$ | $\mathbf{73.17} \pm 0.19$ |
| | $3 \rightarrow 4$ | 11 | 20.19M | $\mathbf{96.05} \pm 0.09$ | $90.33 \pm 0.26$ | $78.70 \pm 0.39$ | $72.75 \pm 0.09$ |
| | $4 \rightarrow 5$ | 11 | 20.19M | $95.88 \pm 0.18$ | $90.26 \pm 0.17$ | $78.12 \pm 0.20$ | $72.28 \pm 0.17$ |
| | $9 \rightarrow 10$ | 11 | 20.19M | $95.96 \pm 0.24$ | $91.08 \pm 0.25$ | $\mathbf{79.33} \pm 0.34$ | $72.00 \pm 0.09$ |
| | - | 12 | 21.82M | $\underline{96.03} \pm 0.24$ | $\underline{90.83} \pm 0.11$ | $\underline{79.06} \pm 0.30$ | $\underline{73.95} \pm 0.09$ |

**Results and Analysis.** As illustrated in Table 1, employing RBA allows for reducing model size while maintaining, and in some cases even improving, performance. Notably, as discussed in Section 4.2 and illustrated in Figure 3 and Figure 5, using `DEiT-S` to approximate the last blocks yields better results, even when approximating multiple blocks such as 9→11 or 8→10. In contrast, with `ViT-S`, the same approximations result in a slight decrease in performance. Moreover, in Figure 6, we illustrate the correlation between the redundancies identified by the BR metric and the results obtained when approximating the identified redundant representations using the `ViT-S` and the `ImageNet1k` dataset. As shown in the leftmost correlation matrix and highlighted in green in the table, approximating redundant blocks yields comparable results while reducing both the number of parameters and computational cost. Additionally, the rightmost correlation matrix, along with the results highlighted in violet in the table, demonstrates that approximating four redundant blocks yields better results than approximating three non-redundant blocks. Overall, performance remains similar or improved, demonstrating that a simple linear transformation is sufficient to approximate different blocks of a NN, significantly reducing the number of parameters and model complexity.

It's important to note that this transformation is uniformly applied to all tokens, further optimizing the process, with no additional training or fine-tuning required afterward. Additional results on classification performance can be found in Table 7.

Table 2: **Image Classification Performance: RBA vs. Skip Across Seeds.** Accuracy scores for `ViT-S` on `CIFAR-10` and `CIFAR-100F` are reported using 3 different seeds. The "Approx." column $b_i \rightarrow b_i + n$ specifies the blocks being approximated, where the first value represents the block whose output is used to approximate the second block's output. The "Skip" column represents the operation of skipping a block instead of approximating it, while the "Num. Blocks" column shows the total number of remaining blocks. Results demonstrate that approximating outperforms skipping in all cases. Refer to Table 9 for results on the other datasets.

| Encoder | Approx. | Num. Blocks | Skip Accuracy ↑ | | Approximate Accuracy ↑ | |
|---|---|---|---|---|---|---|
| | | | CIFAR-10 | CIFAR-100F | CIFAR-10 | CIFAR-100F |
| ViT-S | $1 \rightarrow 5$ | 8 | $58.08 \pm 0.44$ | $32.68 \pm 0.70$ | $84.93 \pm 0.62$ | $58.98 \pm 0.19$ |
| | $2 \rightarrow 5$ | 9 | $64.43 \pm 2.00$ | $41.78 \pm 0.45$ | $90.97 \pm 0.30$ | $69.85 \pm 0.18$ |
| | $7 \rightarrow 10$ | 9 | $73.94 \pm 0.34$ | $45.00 \pm 0.31$ | $85.81 \pm 1.03$ | $60.33 \pm 0.85$ |
| | $1 \rightarrow 3$ | 10 | $66.27 \pm 0.76$ | $42.76 \pm 0.75$ | $92.09 \pm 0.30$ | $72.13 \pm 0.37$ |
| | $2 \rightarrow 4$ | 10 | $71.56 \pm 1.62$ | $50.19 \pm 0.38$ | $93.03 \pm 0.10$ | $74.65 \pm 0.59$ |
| | $9 \rightarrow 11$ | 10 | $89.65 \pm 0.52$ | $70.75 \pm 0.39$ | $89.16 \pm 1.10$ | $68.25 \pm 0.57$ |
| | $2 \rightarrow 3$ | 11 | $81.24 \pm 0.48$ | $60.22 \pm 0.75$ | $94.87 \pm 0.20$ | $79.16 \pm 0.43$ |
| | $9 \rightarrow 10$ | 11 | $93.40 \pm 0.32$ | $76.32 \pm 0.30$ | $94.23 \pm 0.12$ | $76.69 \pm 0.36$ |
| | - | 12 | $95.87 \pm 0.08$ | $81.29 \pm 0.20$ | $95.87 \pm 0.08$ | $81.52 \pm 0.15$ |

**Naive Baseline.** Additionally, we evaluated the model's performance when completely skipping blocks instead of approximating them. As for the previous setting, after performing the desired skips, the network is not trained or fine-tuned. The results in Table 2 show the accuracy scores for `ViT-S` on `CIFAR-10` and `CIFAR-100F`, where the "Skip" column represents the operation of skipping a block entirely rather than applying an approximation. The findings consistently demonstrate that approximating blocks significantly outperforms skipping them in all cases. This underscores the effectiveness of RBA in preserving model performance while reducing complexity. Please refer to Table 9 for results on other datasets.

Table 3: **Generalization Results.** Classification accuracy scores when approximating using a transformation calculated on other datasets for `ViT-S` and `DiNO-S` using `MNIST`, `CIFAR-10`, `CIFAR-100C` and `CIFAR-100F`. The "Approx" column $b_i \rightarrow b_i + n$ specifies the blocks used for approximation, where the first value represents the block whose output is used to approximate the second block's output. The "Fit On" column indicates the dataset on which is calculated the linear transformation. Please refer to Table 10 for complete results.

| Encoder | Approx. | Fit On | Accuracy ↑ | | | |
|---|---|---|---|---|---|---|
| | | | MNIST | CIFAR-10 | CIFAR-100C | CIFAR-100F |
| ViT-S | $2 \rightarrow 3$ | MNIST | 94.11 | 57.13 | 41.89 | 28.50 |
| | | CIFAR-10 | 89.58 | 95.08 | 85.32 | 77.92 |
| | | CIFAR-100 | 89.63 | 95.00 | 85.50 | 77.74 |
| | $3 \rightarrow 4$ | MNIST | 93.52 | 10.36 | 8.97 | 3.09 |
| | | CIFAR-10 | 88.02 | 95.18 | 86.14 | 78.52 |
| | | CIFAR-100 | 88.21 | 94.82 | 85.92 | 78.09 |
| | $1 \rightarrow 3$ | MNIST | 92.79 | 16.17 | 11.09 | 3.84 |
| | | CIFAR-10 | 80.41 | 90.63 | 75.59 | 65.98 |
| | | CIFAR-100 | 81.24 | 89.98 | 76.27 | 66.26 |
| | $3 \rightarrow 5$ | MNIST | 88.22 | 15.17 | 8.52 | 2.03 |
| | | CIFAR-10 | 61.68 | 93.57 | 80.24 | 71.76 |
| | | CIFAR-100 | 64.18 | 92.77 | 80.56 | 72.43 |

**Generalization.** Additionally, we evaluated the model's performance in approximating representations based on a transformation calculated on a different dataset using the same architecture. As in the previous setting, after applying the desired skips, the network is neither trained nor fine-tuned. The results in Table 3 show the accuracy scores for `ViT-S` and `DiNO-S` on `MNIST`, `CIFAR-10`, and `CIFAR-100F`, where the "Fit On" column indicates the dataset used to calculate the transformation. With the exception of `MNIST`, which might be too basic to generalize effectively, the findings

consistently demonstrate that it is possible to leverage a simple linear transformation that is not only shared across all tokens but also across different datasets. Additional results can be found in Table 10.

**Transformation Ablation.** Finally, we conducted an ablation study on the transformations used to approximate latent spaces. The results, presented in Table 4, show accuracy scores for `ViT-S` on `ImageNet1k` using the proposed method (RBA) alongside two more complex MultiLayer Perceptron (MLP) translators, referred to as Res-MLP and MLP. Details on these translators are provided in Appendix A.2.1. Both the MLP and Res-MLP translators are trained for 300 steps using a learning rate of 1e-3 and the Adam optimizer. The findings demonstrate that employing a simple linear transformation to approximate redundant layers is the optimal choice in most cases. As expected, the more blocks are approximated, the less linearly correlated they become, making a more complex approximation more effective (see $1 \rightarrow 5$ in Table 4). Furthermore, the Res-MLP and MLP translators require additional training, whereas the RBA approach is entirely training- and fine-tuning-free, as it relies on a closed-form linear transformation. This process eliminates the need for gradient computation or backpropagation.

Table 4: **Transformation Ablation.** Classification accuracy scores when approximating using RBA or using a more complex MLP on `ImageNet1k` using `ViT-S` accross three seeds. The "Approx" column $b_i \rightarrow b_i + n$ specifies the blocks used for approximation, where the first value represents the block whose output is used to approximate the second block's output.

| Encoder | Approx. | Accuracy ↑ | | |
|---|---|---|---|---|
| | | RBA | MLP | Res-MLP |
| ViT-S | $1 \rightarrow 5$ | $43.68 \pm 0.36$ | $\mathbf{45.79} \pm 0.19$ | $45.44 \pm 0.12$ |
| | $2 \rightarrow 5$ | $\mathbf{60.41} \pm 0.06$ | $60.22 \pm 0.08$ | $60.02 \pm 0.34$ |
| | $7 \rightarrow 10$ | $\mathbf{33.77} \pm 0.44$ | $22.85 \pm 0.10$ | $33.01 \pm 0.76$ |
| | $1 \rightarrow 3$ | $65.31 \pm 0.14$ | $\mathbf{65.45} \pm 0.31$ | $64.54 \pm 0.25$ |
| | $3 \rightarrow 5$ | $\mathbf{68.16} \pm 0.16$ | $66.28 \pm 0.43$ | $67.38 \pm 0.14$ |
| | $2 \rightarrow 4$ | $\mathbf{67.81} \pm 0.15$ | $67.30 \pm 0.12$ | $66.91 \pm 0.09$ |
| | $8 \rightarrow 10$ | $\mathbf{46.75} \pm 0.21$ | $38.29 \pm 0.72$ | $44.97 \pm 0.60$ |
| | $9 \rightarrow 11$ | $\mathbf{46.17} \pm 0.25$ | $34.70 \pm 0.68$ | $39.01 \pm 0.34$ |
| | $2 \rightarrow 3$ | $\mathbf{71.74} \pm 0.29$ | $71.25 \pm 0.19$ | $70.94 \pm 0.18$ |
| | $3 \rightarrow 4$ | $\mathbf{71.70} \pm 0.28$ | $70.78 \pm 0.42$ | $70.78 \pm 0.10$ |
| | $4 \rightarrow 5$ | $\mathbf{71.49} \pm 0.23$ | $69.47 \pm 0.18$ | $70.86 \pm 0.10$ |
| | $9 \rightarrow 10$ | $\mathbf{61.11} \pm 0.15$ | $53.78 \pm 0.19$ | $58.06 \pm 0.43$ |

**Takeaway.** Redundant Block Approximation preserves essential representational features while maintaining the model's structural integrity, even when simplifying its architecture, whereas just skipping blocks could lead to performance degradation.

## 5 CONCLUSION

In this paper, we introduced a novel framework for approximating redundant representations in transformer-based foundation models and proposed a simple yet effective metric to identify such redundancies. By leveraging a simple linear transformation, shared across all tokens, between consecutive and non-consecutive blocks output, we demonstrated that it is possible to significantly reduce model parameters and complexity without sacrificing performance, and in some cases even improving it. Our approach provides an efficient way to optimize model architecture, maintaining essential representation fidelity while streamlining the network for downstream tasks.

**Limitations and Future Works.** While our framework shows promising results, it has been primarily tested on transformer-based architectures. We leave to future work to explore the application of our framework across different modalities (e.g., text), architectures (e.g., ResNets and AutoEncoders), and downstream tasks (e.g., reconstruction). Additionally, we plan to enhance the representation analysis by incorporating topological metrics, which could provide a different perspective on structural similarities between representations. This alternative viewpoint may uncover new insights into redundancy patterns and further refine our approach. By expanding the framework's scope, we aim to validate its versatility and continue optimizing model efficiency across a broader set of architectures and tasks, advancing its practical applicability in diverse settings.

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

# A APPENDIX

## A.1 REPRODUCIBILITY STATEMENT

In Section Section 3, we provide a detailed description of the proposed framework and the experimental settings for the various scenarios. In the following sections, we present all implementation details that are not described in the main manuscript. Additionally, we will release a modular PyTorch implementation.

## A.2 IMPLEMENTATION DETAILS

This section details the experiments conducted in Section 4. Table 5 contains the full list of the pretrained models, while Table 6 contains dataset information.

Table 5: **Pretrained models details.** Details of the pretrained feature extractors with their Hugging-Face key, their alias, and their latent space dimensionality.

| Modality | HuggingFace Model Name | Alias | Enc. Dim |
|---|---|---|---|
| Vision | WinKawaks/vit-small-patch16-224 | ViT-S (Dosovitskiy et al., 2021) | 384 |
| | google/vit-base-patch16-224 | ViT-B (Dosovitskiy et al., 2021) | 768 |
| | facebook/dinov2-small | DiNO-S (Oquab et al., 2023) | 384 |
| | facebook/dinov2-base | DiNO-B (Oquab et al., 2023) | 768 |
| | facebook/deit-small-patch16-224 | DEiT-S (Touvron et al., 2020) | 384 |

Table 6: **Dataset details.** Details of the HuggingFace datasets used in the classification and reconstruction experiments, with the associated number of classes.

| Modality | Name | Alias | Number of Classes |
|---|---|---|---|
| Image | MNIST (Deng, 2012) | MNIST | 10 |
| | Fasion-MNIST (Xiao et al., 2017) | F-MNIST | 10 |
| | CIFAR-10 (Krizhevsky et al., 2009) | CIFAR-10 | 10 |
| | CIFAR-100 (coarse) (Krizhevsky et al., 2009) | CIFAR-100C | 20 |
| | CIFAR-100 (fine) (Krizhevsky et al., 2009) | CIFAR-100F | 100 |
| | Imagenet-1k (Russakovsky et al., 2015) | ImageNet1k | 1000 |

### A.2.1 TRANSLATORS

The first implementation, referred to as the Res-MLP, is composed of two normalization layers and a feedforward submodule. The first layer normalization processes the input tensor, followed by a feedforward submodule comprising a linear transformation, a SiLU activation, a dropout layer, and a final linear transformation. The output of the feedforward submodule is added to the normalized input via a residual connection. This sum is then passed through the second layer normalization to produce the final output. While the second implementation, referred to as the MLP, is a simplified MLP that employs a sequential architecture with a first linear transformation that reduces the input dimensionality to half of the target dimension, followed by a GELU activation function for smooth non-linearity, and a final linear transformation that restores the reduced features to match the target dimensionality. Refer to Listings 1 and 2 for the code snipped of the two translators.

Listing 1: Python Code Snippet for the Res-MLP translator

```python
class ResMLP(nn.Module):
    def __init__(self, num_features: int, dropout_p: float):
        super().__init__()

        self.norm1 = nn.LayerNorm(num_features)
        self.norm2 = nn.LayerNorm(num_features)

        self.ff = nn.Sequential(
```

```
            nn.Linear(num_features, num_features),
            nn.SiLU(),
            nn.Dropout(p=dropout_p),
            nn.Linear(num_features, num_features),
        )

    def forward(self, x: torch.Tensor) -> torch.Tensor:
        x_normalized = self.norm1(x)
        x_transformed = self.ff(x_normalized)
        return self.norm2(x_transformed + x_normalized)
```

Listing 2: Python Code Snippet for the MLP translator

```
translation = nn.Sequential(
    nn.Linear(x.size(1), y.size(1) // 2),
    nn.GELU(),
    nn.Linear(y.size(1) // 2, y.size(1)),
)
```

### A.2.2 TOOLS & TECHNOLOGIES

All the experiments presented in this work employ the following tools:

- *PyTorch Lightning*, to ensure reproducible results while also getting a clean and modular codebase;
- *NN-Template GrokAI (2021)*, to easily bootstrap the project and enforce best practices;
- *Transformers by HuggingFace*, to get ready-to-use transformers for both text and images;
- *Datasets by HuggingFace*, to access most of the datasets;
- *DVC* (Kuprieiev et al., 2022), for data versioning;

### A.3 ADDITIONAL EXPERIMENTS

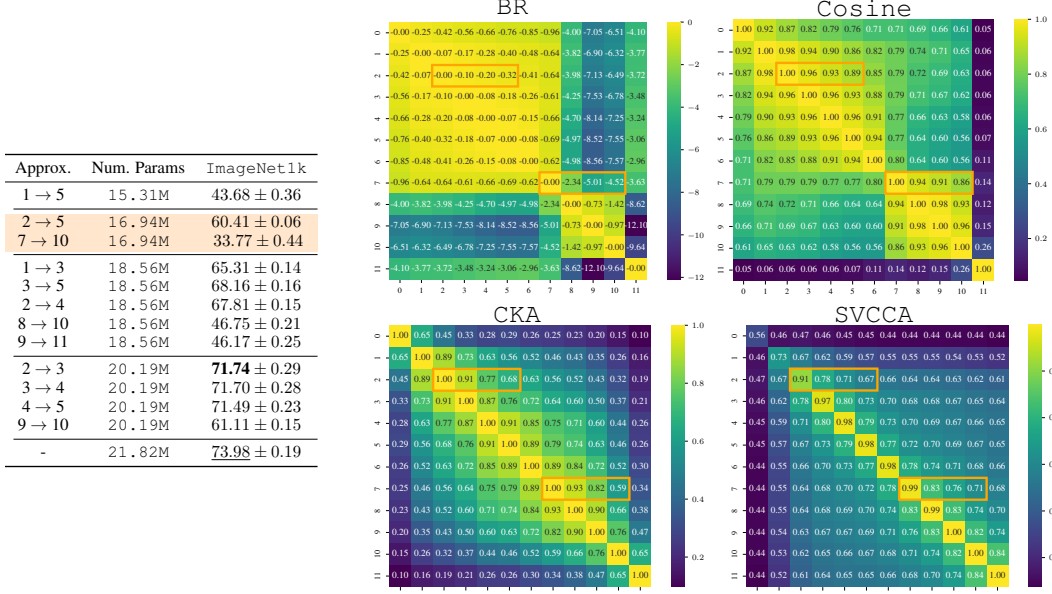

| Approx. | Num. Params | ImageNet1k |
|---|---|---|
| $1 \rightarrow 5$ | 15.31M | $43.68 \pm 0.36$ |
| $2 \rightarrow 5$ | 16.94M | $60.41 \pm 0.06$ |
| $7 \rightarrow 10$ | 16.94M | $33.77 \pm 0.44$ |
| $1 \rightarrow 3$ | 18.56M | $65.31 \pm 0.14$ |
| $3 \rightarrow 5$ | 18.56M | $68.16 \pm 0.16$ |
| $2 \rightarrow 4$ | 18.56M | $67.81 \pm 0.15$ |
| $8 \rightarrow 10$ | 18.56M | $46.75 \pm 0.21$ |
| $9 \rightarrow 11$ | 18.56M | $46.17 \pm 0.25$ |
| $2 \rightarrow 3$ | 20.19M | $\mathbf{71.74} \pm 0.29$ |
| $3 \rightarrow 4$ | 20.19M | $71.70 \pm 0.28$ |
| $4 \rightarrow 5$ | 20.19M | $71.49 \pm 0.23$ |
| $9 \rightarrow 10$ | 20.19M | $61.11 \pm 0.15$ |
| - | 21.82M | $\underline{73.98} \pm 0.19$ |

Figure 7: **Correlation Between Similarity Metrics and Accuracy Approximation.** (*Left*) The accuracy performance of the `ViT-S` encoder is shown with different approximation strategies applied on `ImageNet1k`. (*Right*) A block-by-block matrix visualizes results using various similarity metrics. The findings reveal that using BR or cosine similarity enhances the emergence of the block structure, making it easier to identify highly redundant blocks. Notably, results highlighted in *orange* demonstrate that the BR metric uniquely indicates that approximating blocks $7 \rightarrow 10$ is suboptimal, as it results in lower accuracy. In contrast, approximating an equivalent number of blocks (3) in the range of $2 \rightarrow 5$ yields favorable results, particularly when considering the reduction in the number of parameters.

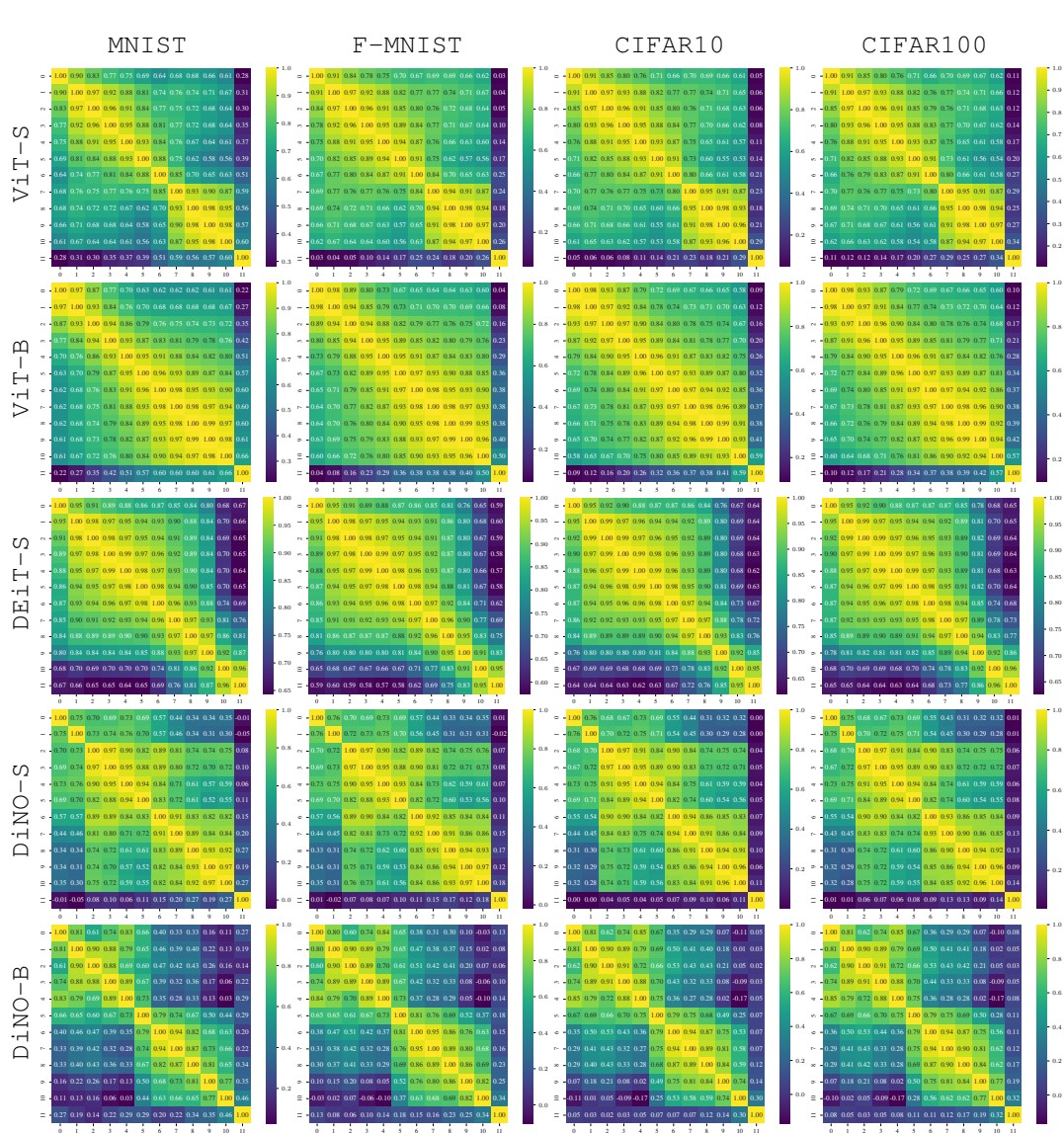

Figure 8: **Representation Similarities.** Cosine similarity matrices illustrating the internal block-by-block similarities in `ViT-S`, `ViT-B`, `DEiT-S`, `DiNO-S` and `DiNO-B`, and `DEiT-S` models across four datasets: `MNIST`, `F-MNIST`, `CIFAR-10`, and `CIFAR-100`. Each heatmap quantifies the similarity between the internal representations of different blocks using the [CLS] token, providing insights into redundancy in foundation pretrained models. The matrices reveal that the similarity structure between computational blocks is predominantly influenced by the model architecture itself rather than the specific dataset.

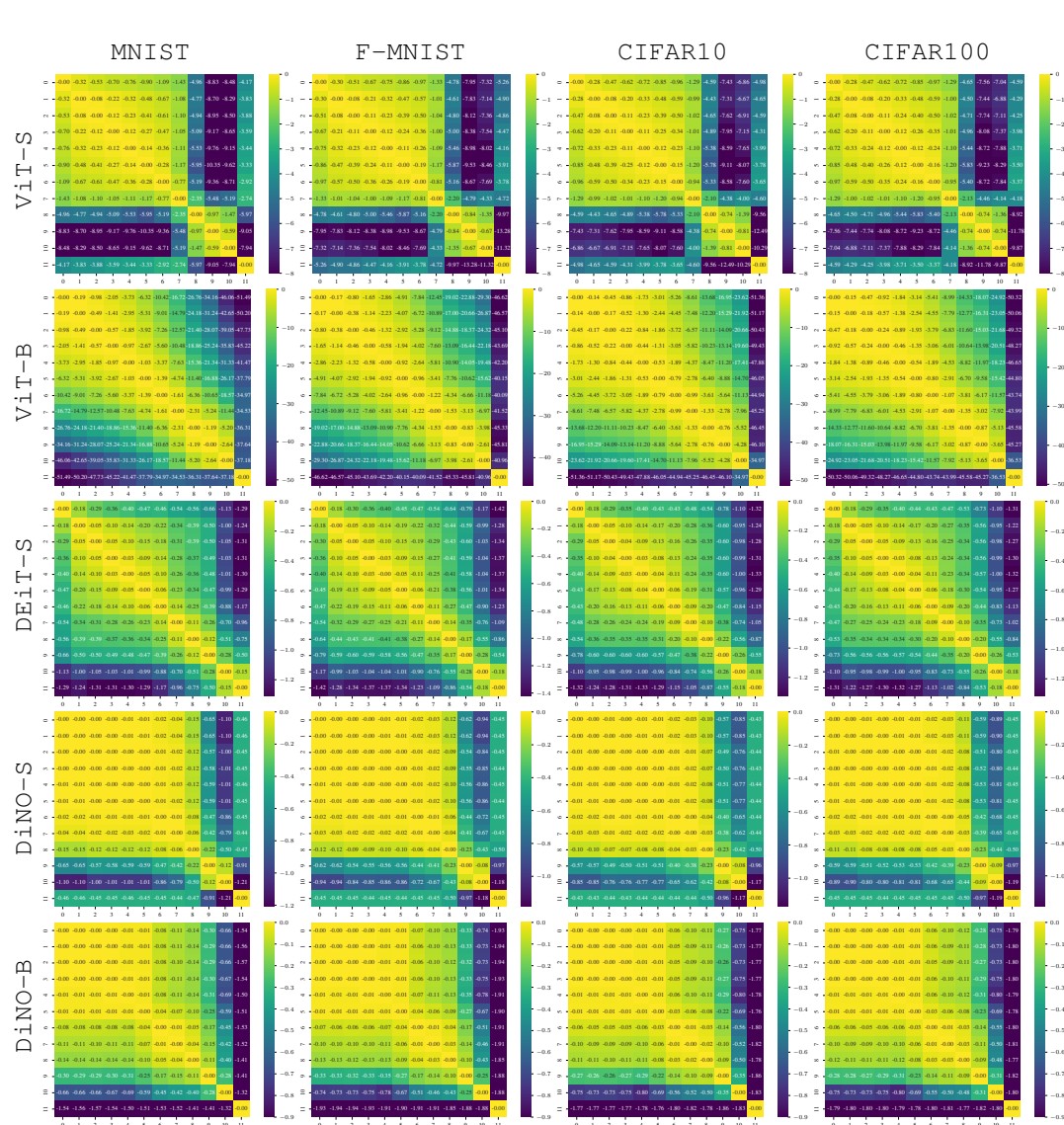

Figure 9: **Representation Redundancies.** BR matrices illustrating the internal block-by-block redundancies in `ViT-S`, `ViT-B`, `DEiT-S`, `DiNO-S` and `DiNO-B`, and `DEiT-S` models across four datasets: `MNIST`, `F-MNIST`, `CIFAR-10`, and `CIFAR-100`. Each heatmap quantifies the BR metric between the internal representations of different blocks using the [CLS] token, providing insights into redundancy in foundation pretrained models. The matrices reveal that the similarity structure between computational blocks is predominantly influenced by the model architecture itself rather than the specific dataset.

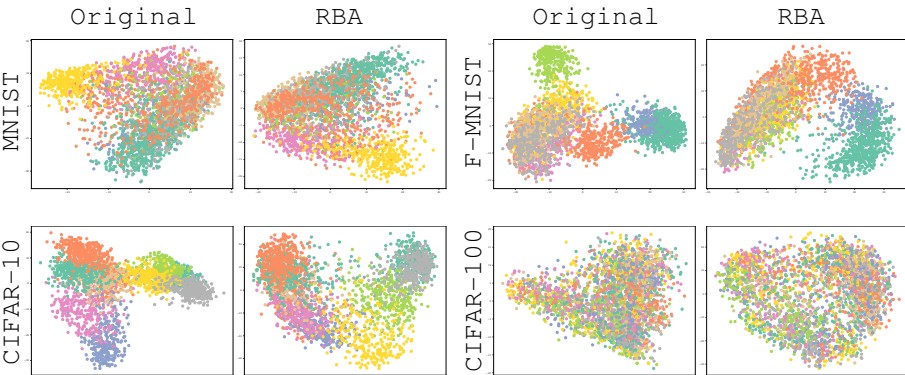

Figure 10: **Last Block Approximation.** PCA visualization of the final layer representations for both the original model and the model with its second block approximated by the preceding one. The representations are generated using the `DiNO-S` model across four datasets. Note that for `CIFAR-100` (bottom right), only the overall structure of the space can be observed, as the 100 classes make it challenging to distinguish labels based on color

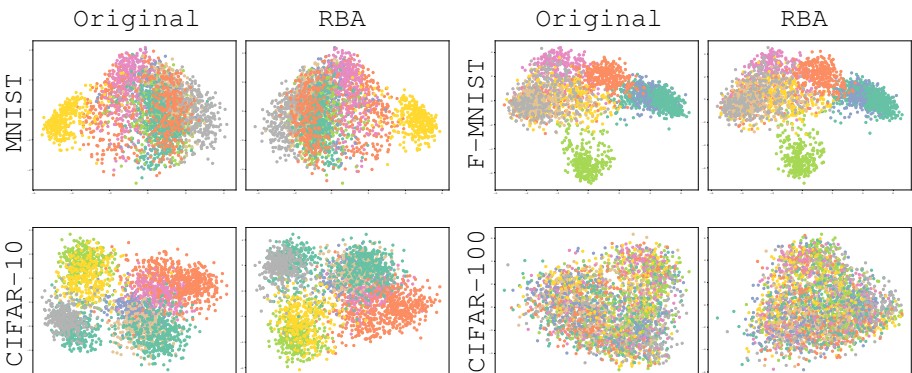

Figure 11: **Last Block Approximation.** PCA visualization of the last layer representations for both the original model and the model with its second block approximated using the previous one. Representations refer to the using `ViT-S` model across four datasets.

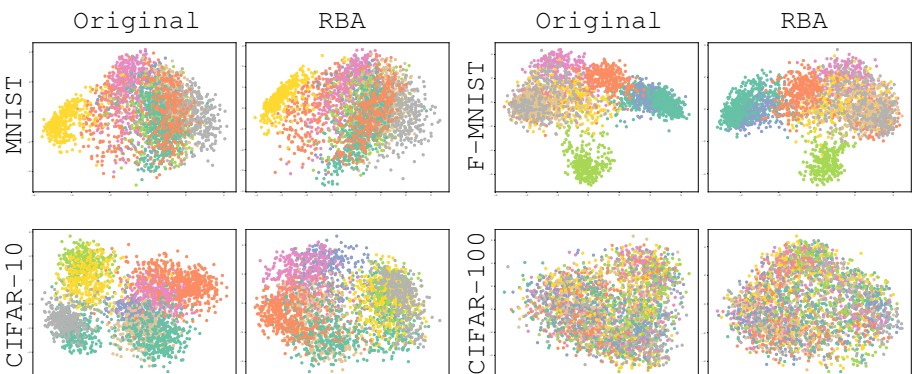

Figure 12: **Last Block Approximation.** PCA visualization of the last layer representations for both the original model and the model with its last block approximated from the previous one. Representations refer to the using `ViT-S` model across four datasets.

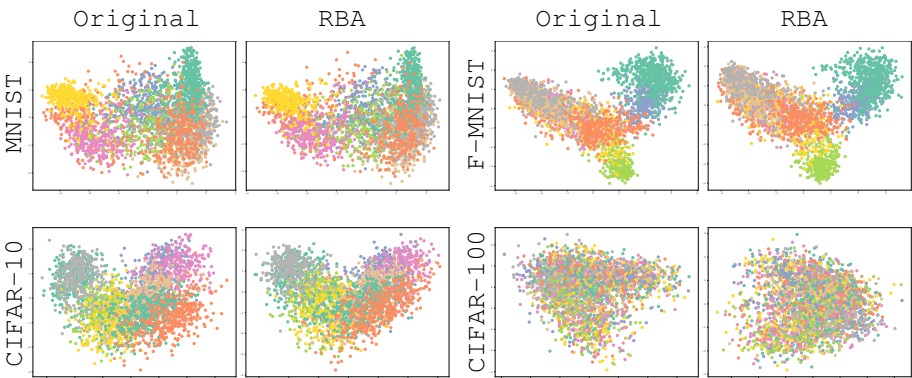

Figure 13: **Last Block Approximation.** PCA visualization of the last layer representations for both the original model and the model with its last block approximated from the previous one. Representations refer to the using `DEiT-S` model across four datasets.

Table 7: **Image Classification Performance Across Architectures and Seeds.** Accuracy scores are reported for different pretrained models, random seeds, and datasets. CIFAR-100C refers to CIFAR-100 with the coarse setting (20 labels), while CIFAR-100F refers to the fine setting (100 labels). The "Approx" column $b_i \rightarrow b_i + n$ specifies the blocks used for approximation, where the first value represents the block whose output is used to approximate the second block's output. The "Num. Blocks" column indicates the total number of remaining blocks after the approximation, and the "Num. Params" column shows the number of model parameters. The proposed method preserves performance while reducing the number of parameters.

| Encoder | Approx. | Num. Blocks | Num. Params | Accuracy ↑ | | | | |
| | | | | MNIST | F-MNIST | CIFAR-10 | CIFAR-100C | CIFAR-100F |
|---|---|---|---|---|---|---|---|---|
| ViT-S | $1 \rightarrow 5$ | 8 | 15.31M | $92.11 \pm 0.20$ | $86.36 \pm 1.00$ | $84.93 \pm 0.62$ | $68.47 \pm 0.30$ | $58.96 \pm 0.20$ |
| | $2 \rightarrow 5$ | 9 | 16.94M | $94.67 \pm 0.12$ | $87.82 \pm 0.92$ | $90.97 \pm 0.30$ | $78.07 \pm 0.38$ | $69.83 \pm 0.19$ |
| | $7 \rightarrow 10$ | 9 | 16.94M | $94.91 \pm 0.30$ | $88.00 \pm 0.78$ | $85.81 \pm 1.03$ | $71.10 \pm 0.51$ | $60.18 \pm 0.93$ |
| | $1 \rightarrow 3$ | 10 | 18.56M | $95.67 \pm 0.19$ | $87.43 \pm 0.63$ | $92.09 \pm 0.30$ | $79.68 \pm 0.20$ | $72.12 \pm 0.27$ |
| | $3 \rightarrow 5$ | 10 | 18.56M | $95.16 \pm 0.08$ | $88.38 \pm 0.80$ | $94.18 \pm 0.11$ | $83.29 \pm 0.47$ | $76.46 \pm 0.23$ |
| | $2 \rightarrow 4$ | 10 | 18.56M | $95.37 \pm 0.08$ | $88.08 \pm 1.08$ | $93.03 \pm 0.10$ | $81.74 \pm 0.28$ | $74.69 \pm 0.60$ |
| | $8 \rightarrow 10$ | 10 | 18.56M | $95.27 \pm 0.58$ | $88.56 \pm 0.95$ | $91.56 \pm 0.72$ | $77.73 \pm 0.41$ | $69.36 \pm 0.22$ |
| | $9 \rightarrow 11$ | 10 | 18.56M | $94.77 \pm 0.10$ | $88.23 \pm 0.42$ | $89.16 \pm 1.10$ | $75.30 \pm 0.44$ | $68.19 \pm 0.59$ |
| | $2 \rightarrow 3$ | 11 | 20.19M | $\mathbf{95.76} \pm 0.08$ | $88.67 \pm 0.63$ | $94.87 \pm 0.20$ | $85.96 \pm 0.05$ | $79.21 \pm 0.45$ |
| | $3 \rightarrow 4$ | 11 | 20.19M | $95.70 \pm 0.11$ | $88.35 \pm 1.00$ | $95.10 \pm 0.23$ | $86.00 \pm 0.12$ | $79.57 \pm 0.43$ |
| | $4 \rightarrow 5$ | 11 | 20.19M | $95.67 \pm 0.17$ | $89.11 \pm 0.45$ | $\mathbf{95.43} \pm 0.25$ | $\mathbf{86.24} \pm 0.21$ | $\mathbf{79.87} \pm 0.20$ |
| | $9 \rightarrow 10$ | 11 | 20.19M | $95.75 \pm 0.44$ | $\mathbf{88.85} \pm 0.90$ | $94.23 \pm 0.12$ | $82.69 \pm 0.49$ | $76.65 \pm 0.37$ |
| | - | 12 | 21.82M | $\underline{95.95} \pm 0.40$ | $\underline{89.01} \pm 0.63$ | $\underline{95.87} \pm 0.08$ | $\underline{87.60} \pm 0.15$ | $\underline{81.44} \pm 0.19$ |
| DiNO-S | $1 \rightarrow 5$ | 8 | 15.55M | $95.32 \pm 1.09$ | $87.43 \pm 0.78$ | $79.37 \pm 1.34$ | $60.72 \pm 0.49$ | $51.72 \pm 0.44$ |
| | $2 \rightarrow 5$ | 9 | 17.18M | $96.04 \pm 0.67$ | $88.43 \pm 0.65$ | $85.58 \pm 0.54$ | $67.89 \pm 0.57$ | $60.21 \pm 0.60$ |
| | $7 \rightarrow 10$ | 9 | 17.18M | $96.93 \pm 0.45$ | $87.47 \pm 0.74$ | $91.24 \pm 0.13$ | $78.14 \pm 0.14$ | $70.46 \pm 0.23$ |
| | $1 \rightarrow 3$ | 10 | 18.80M | $96.74 \pm 0.96$ | $87.60 \pm 1.68$ | $91.82 \pm 0.17$ | $78.81 \pm 0.35$ | $71.79 \pm 0.22$ |
| | $3 \rightarrow 5$ | 10 | 18.80M | $96.93 \pm 0.42$ | $88.54 \pm 0.21$ | $90.90 \pm 0.30$ | $76.12 \pm 0.50$ | $69.16 \pm 0.74$ |
| | $2 \rightarrow 4$ | 10 | 18.80M | $96.54 \pm 0.55$ | $87.63 \pm 1.29$ | $91.03 \pm 0.75$ | $76.57 \pm 0.25$ | $69.82 \pm 0.60$ |
| | $8 \rightarrow 10$ | 10 | 18.80M | $97.03 \pm 0.17$ | $87.77 \pm 1.38$ | $93.34 \pm 0.44$ | $82.27 \pm 0.41$ | $75.02 \pm 1.12$ |
| | $9 \rightarrow 11$ | 10 | 18.80M | $92.46 \pm 1.63$ | $82.68 \pm 0.92$ | $85.65 \pm 0.68$ | $72.44 \pm 1.19$ | $60.73 \pm 0.62$ |
| | $2 \rightarrow 3$ | 11 | 20.43M | $96.99 \pm 0.70$ | $\mathbf{88.62} \pm 0.54$ | $94.67 \pm 0.20$ | $83.92 \pm 0.49$ | $\mathbf{78.34} \pm 0.30$ |
| | $3 \rightarrow 4$ | 11 | 20.43M | $97.22 \pm 0.50$ | $88.06 \pm 1.01$ | $\mathbf{94.72} \pm 0.24$ | $83.37 \pm 0.37$ | $78.14 \pm 0.20$ |
| | $4 \rightarrow 5$ | 11 | 20.43M | $\mathbf{97.33} \pm 0.47$ | $88.67 \pm 1.36$ | $94.64 \pm 0.10$ | $82.81 \pm 0.62$ | $76.99 \pm 0.37$ |
| | $9 \rightarrow 10$ | 11 | 20.43M | $96.99 \pm 0.97$ | $88.41 \pm 0.33$ | $93.52 \pm 0.48$ | $\mathbf{84.09} \pm 0.52$ | $77.54 \pm 0.89$ |
| | - | 12 | 22.06M | $\underline{96.85} \pm 1.04$ | $\underline{88.17} \pm 0.64$ | $96.06 \pm 0.32$ | $\underline{87.62} \pm 0.24$ | $\underline{82.09} \pm 0.23$ |
| DEiT-S | $1 \rightarrow 5$ | 8 | 15.31M | $93.27 \pm 0.37$ | $85.76 \pm 0.30$ | $78.20 \pm 0.21$ | $59.82 \pm 0.16$ | $50.72 \pm 0.31$ |
| | $2 \rightarrow 5$ | 9 | 16.94M | $94.99 \pm 0.18$ | $87.41 \pm 0.27$ | $85.27 \pm 0.11$ | $69.95 \pm 0.15$ | $61.25 \pm 0.29$ |
| | $7 \rightarrow 10$ | 9 | 16.94M | $95.81 \pm 0.23$ | $87.82 \pm 0.43$ | $89.20 \pm 0.34$ | $75.96 \pm 0.20$ | $69.22 \pm 0.21$ |
| | $1 \rightarrow 3$ | 10 | 18.56M | $95.35 \pm 0.21$ | $87.11 \pm 0.32$ | $85.59 \pm 0.23$ | $70.61 \pm 0.42$ | $61.74 \pm 0.07$ |
| | $3 \rightarrow 5$ | 10 | 18.56M | $95.86 \pm 0.14$ | $87.79 \pm 0.51$ | $89.12 \pm 0.23$ | $75.84 \pm 0.09$ | $67.25 \pm 0.20$ |
| | $2 \rightarrow 4$ | 10 | 18.56M | $95.68 \pm 0.11$ | $87.96 \pm 0.39$ | $88.76 \pm 0.08$ | $75.83 \pm 0.38$ | $67.01 \pm 0.31$ |
| | $8 \rightarrow 10$ | 10 | 18.56M | $95.87 \pm 0.27$ | $88.05 \pm 0.37$ | $90.62 \pm 0.09$ | $78.25 \pm 0.52$ | $71.03 \pm 0.31$ |
| | $9 \rightarrow 11$ | 10 | 18.56M | $95.64 \pm 0.13$ | $\mathbf{88.26} \pm 0.11$ | $91.09 \pm 0.21$ | $79.30 \pm 0.58$ | $\mathbf{71.77} \pm 0.33$ |
| | $2 \rightarrow 3$ | 11 | 20.19M | $95.99 \pm 0.19$ | $87.85 \pm 0.33$ | $90.13 \pm 0.23$ | $78.11 \pm 0.23$ | $70.13 \pm 0.09$ |
| | $3 \rightarrow 4$ | 11 | 20.19M | $\mathbf{96.05} \pm 0.09$ | $87.97 \pm 0.14$ | $90.33 \pm 0.26$ | $78.70 \pm 0.39$ | $70.40 \pm 0.21$ |
| | $4 \rightarrow 5$ | 11 | 20.19M | $95.88 \pm 0.18$ | $88.04 \pm 0.31$ | $90.26 \pm 0.17$ | $78.12 \pm 0.20$ | $69.66 \pm 0.38$ |
| | $9 \rightarrow 10$ | 11 | 20.19M | $95.96 \pm 0.24$ | $88.09 \pm 0.17$ | $91.08 \pm 0.25$ | $\mathbf{79.33} \pm 0.34$ | $71.62 \pm 0.10$ |
| | - | 12 | 21.82M | $\underline{96.03} \pm 0.24$ | $\underline{87.86} \pm 0.25$ | $\underline{90.83} \pm 0.11$ | $\underline{79.06} \pm 0.30$ | $\underline{71.25} \pm 0.18$ |

Table 8: **Image Classification Performance Across Seeds.** Accuracy scores are reported for `ViT-B` using 3 random seeds, and different datasets. `CIFAR-100C` refers to `CIFAR-100` with the `coarse` setting (20 labels), while `CIFAR-100F` refers to the `fine` setting (100 labels). The "Approx." column $b_i \rightarrow b_i + n$ specify the blocks used for approximation, where the first value represents the block whose output is used to approximate the second block's output, while the "Num. Blocks" column indicates the total number of remaining blocks after the approximation. The proposed method preserves performance while reducing the number of parameters.

| | | Accuracy ↑ | | | | |
|---|---|---|---|---|---|---|
| Approx. | Num. Params | MNIST | F-MNIST | CIFAR-10 | CIFAR-100C | CIFAR-100F |
| $1 \rightarrow 5$ | 60.40M | $87.06 \pm 0.53$ | $84.33 \pm 0.61$ | $73.54 \pm 0.57$ | $51.67 \pm 1.10$ | $38.98 \pm 0.72$ |
| $2 \rightarrow 5$ | 66.90M | $94.20 \pm 0.21$ | $87.80 \pm 0.24$ | $87.10 \pm 0.83$ | $71.68 \pm 0.50$ | $61.19 \pm 0.37$ |
| $1 \rightarrow 3$ | 73.40M | $96.51 \pm 0.42$ | $88.72 \pm 0.41$ | $93.71 \pm 0.13$ | $83.05 \pm 0.23$ | $74.74 \pm 0.29$ |
| $3 \rightarrow 5$ | 73.40M | $95.59 \pm 0.09$ | $88.28 \pm 0.20$ | $93.11 \pm 0.06$ | $83.50 \pm 0.17$ | $74.35 \pm 0.47$ |
| $2 \rightarrow 4$ | 73.40M | $96.21 \pm 0.33$ | $89.21 \pm 0.64$ | $94.59 \pm 0.32$ | $85.13 \pm 0.24$ | $76.82 \pm 0.41$ |
| $8 \rightarrow 10$ | 73.40M | $96.54 \pm 0.21$ | $\mathbf{89.72} \pm 0.52$ | $95.05 \pm 0.26$ | $85.78 \pm 0.37$ | $79.62 \pm 0.14$ |
| $9 \rightarrow 11$ | 73.40M | $95.59 \pm 0.52$ | $89.49 \pm 0.26$ | $93.22 \pm 0.56$ | $82.23 \pm 0.44$ | $76.33 \pm 0.10$ |
| $3 \rightarrow 4$ | 79.90M | $96.86 \pm 0.35$ | $89.69 \pm 1.09$ | $\mathbf{96.18} \pm 0.09$ | $\mathbf{89.18} \pm 0.06$ | $\mathbf{82.50} \pm 0.17$ |
| $4 \rightarrow 5$ | 79.90M | $96.55 \pm 0.23$ | $89.13 \pm 0.50$ | $95.39 \pm 0.23$ | $87.43 \pm 0.15$ | $80.30 \pm 0.16$ |
| $0 \rightarrow 1$ | 79.90M | $96.75 \pm 0.29$ | $88.97 \pm 0.26$ | $93.74 \pm 0.15$ | $84.49 \pm 0.20$ | $76.54 \pm 0.29$ |
| $1 \rightarrow 2$ | 79.90M | $96.88 \pm 0.01$ | $89.29 \pm 0.24$ | $95.63 \pm 0.11$ | $87.46 \pm 0.20$ | $80.64 \pm 0.23$ |
| $2 \rightarrow 3$ | 79.90M | $\mathbf{96.91} \pm 0.17$ | $89.69 \pm 0.61$ | $96.00 \pm 0.18$ | $88.38 \pm 0.13$ | $81.59 \pm 0.35$ |
| - | 86.40M | $\underline{95.61} \pm 0.22$ | $\underline{89.64} \pm 0.57$ | $\underline{96.25} \pm 0.17$ | $\underline{89.52} \pm 0.23$ | $\underline{83.41} \pm 0.20$ |

Table 9: **Image Classification Performance: RBA vs. Skip Across Seeds.** Accuracy scores for `ViT-S` on all the datasets are reported using 3 different seeds. The "Skip." column $b_i \rightarrow b_i + n$ specifies the blocks being skipped, where the first value represents the starting block (excluded from the skip) and the second value represents the final (included) block. The "Num. Blocks" column shows the total number of remaining blocks.

| | | Skip Accuracy ↑ | | | | |
|---|---|---|---|---|---|---|
| Skip | Num. Blocks | MNIST | F-MNIST | CIFAR-10 | CIFAR-100C | CIFAR-100F |
| $1 \rightarrow 5$ | 8 | $92.74 \pm 0.58$ | $82.25 \pm 0.93$ | $58.08 \pm 0.44$ | $43.43 \pm 0.79$ | $32.68 \pm 0.70$ |
| $2 \rightarrow 5$ | 9 | $93.78 \pm 0.55$ | $84.99 \pm 0.51$ | $64.43 \pm 2.00$ | $51.39 \pm 0.57$ | $41.78 \pm 0.45$ |
| $7 \rightarrow 10$ | 9 | $91.56 \pm 0.46$ | $85.02 \pm 1.15$ | $73.94 \pm 0.34$ | $59.99 \pm 0.73$ | $45.00 \pm 0.31$ |
| $1 \rightarrow 3$ | 10 | $94.41 \pm 0.33$ | $82.82 \pm 0.46$ | $66.27 \pm 0.76$ | $52.52 \pm 0.48$ | $42.76 \pm 0.75$ |
| $3 \rightarrow 5$ | 10 | $93.96 \pm 0.25$ | $86.10 \pm 0.15$ | $74.79 \pm 1.56$ | $62.53 \pm 0.32$ | $54.62 \pm 0.52$ |
| $2 \rightarrow 4$ | 10 | $94.31 \pm 0.48$ | $85.22 \pm 0.67$ | $71.56 \pm 1.62$ | $59.40 \pm 0.38$ | $50.19 \pm 0.38$ |
| $8 \rightarrow 10$ | 10 | $94.82 \pm 0.21$ | $87.77 \pm 0.43$ | $85.74 \pm 0.32$ | $72.39 \pm 0.41$ | $63.79 \pm 0.66$ |
| $9 \rightarrow 11$ | 10 | $94.80 \pm 0.15$ | $88.32 \pm 0.46$ | $89.65 \pm 0.52$ | $76.40 \pm 0.08$ | $70.75 \pm 0.39$ |
| $0 \rightarrow 1$ | 11 | $95.98 \pm 0.13$ | $84.91 \pm 0.36$ | $70.90 \pm 0.09$ | $57.16 \pm 0.41$ | $47.54 \pm 0.37$ |
| $1 \rightarrow 2$ | 11 | $95.79 \pm 0.16$ | $87.07 \pm 0.70$ | $83.21 \pm 0.52$ | $70.66 \pm 0.69$ | $62.23 \pm 0.21$ |
| $2 \rightarrow 3$ | 11 | $95.14 \pm 0.39$ | $85.50 \pm 0.62$ | $81.24 \pm 0.48$ | $68.63 \pm 0.33$ | $60.22 \pm 0.75$ |
| $3 \rightarrow 4$ | 11 | $95.34 \pm 0.58$ | $87.62 \pm 1.18$ | $88.25 \pm 0.23$ | $77.58 \pm 0.46$ | $69.79 \pm 0.02$ |
| $4 \rightarrow 5$ | 11 | $95.75 \pm 0.20$ | $87.26 \pm 0.86$ | $86.23 \pm 0.63$ | $74.52 \pm 0.63$ | $66.69 \pm 0.48$ |
| $5 \rightarrow 6$ | 11 | $95.77 \pm 0.22$ | $86.99 \pm 0.33$ | $83.42 \pm 0.52$ | $69.62 \pm 0.32$ | $61.96 \pm 0.55$ |
| $6 \rightarrow 7$ | 11 | $95.33 \pm 0.08$ | $86.64 \pm 1.14$ | $87.57 \pm 0.24$ | $75.91 \pm 0.20$ | $68.70 \pm 0.31$ |
| $7 \rightarrow 8$ | 11 | $95.76 \pm 0.20$ | $87.50 \pm 0.85$ | $88.70 \pm 0.46$ | $76.80 \pm 0.09$ | $69.33 \pm 0.39$ |
| $8 \rightarrow 9$ | 11 | $96.28 \pm 0.04$ | $88.38 \pm 0.83$ | $89.98 \pm 0.48$ | $76.45 \pm 0.65$ | $71.80 \pm 0.22$ |
| $9 \rightarrow 10$ | 11 | $95.56 \pm 0.47$ | $88.74 \pm 1.09$ | $93.40 \pm 0.32$ | $82.44 \pm 0.44$ | $76.32 \pm 0.30$ |
| $10 \rightarrow 11$ | 11 | $95.22 \pm 0.29$ | $89.39 \pm 0.30$ | $93.77 \pm 0.69$ | $82.39 \pm 0.06$ | $78.68 \pm 0.29$ |
| - | 12 | $95.95 \pm 0.40$ | $89.01 \pm 0.63$ | $95.87 \pm 0.08$ | $87.60 \pm 0.15$ | $81.29 \pm 0.20$ |

Table 10: **Generalization Results.** Classification accuracy scores when approximating using a transformation calculated on other datasets for `ViT-S` and `DiNO-S` using `MNIST`, `CIFAR-10`, `CIFAR-100C` and `CIFAR-100F`. `CIFAR-100C` refers to `CIFAR-100` with the `coarse` setting (20 labels), while `CIFAR-100F` with the `fine` setting (100 labels). The "Approx" column $b_i \to b_i + n$ specifies the blocks used for approximation, where the first value represents the block whose output is used to approximate the second block's output. The "Fit on" column indicates the dataset on which the linear transformation is calculated.

| Encoder | Approx. | Fit On | Accuracy ↑ | | | |
|---|---|---|---|---|---|---|
| | | | MNIST | CIFAR-10 | CIFAR-100C | CIFAR-100F |
| ViT-S | 2 → 3 | MNIST | 94.11 | 57.13 | 41.89 | 28.50 |
| | | CIFAR-10 | 89.58 | 95.08 | 85.32 | 77.92 |
| | | CIFAR-100 | 89.63 | 95.00 | 85.50 | 77.74 |
| | 3 → 4 | MNIST | 93.52 | 10.36 | 8.97 | 3.09 |
| | | CIFAR-10 | 88.02 | 95.18 | 86.14 | 78.52 |
| | | CIFAR-100 | 88.21 | 94.82 | 85.92 | 78.09 |
| | 4 → 5 | MNIST | 93.96 | 38.40 | 25.56 | 16.52 |
| | | CIFAR-10 | 78.36 | 95.31 | 85.84 | 78.20 |
| | | CIFAR-100 | 80.11 | 94.98 | 86.01 | 78.14 |
| | 9 → 10 | MNIST | 89.73 | 74.41 | 59.78 | 44.40 |
| | | CIFAR-10 | 82.28 | 92.39 | 71.63 | 57.17 |
| | | CIFAR-100 | 54.12 | 85.60 | 77.37 | 61.81 |
| | 1 → 3 | MNIST | 92.79 | 16.17 | 11.09 | 3.84 |
| | | CIFAR-10 | 80.41 | 90.63 | 75.59 | 65.98 |
| | | CIFAR-100 | 81.24 | 89.98 | 76.27 | 66.26 |
| | 3 → 5 | MNIST | 88.22 | 15.17 | 8.52 | 2.03 |
| | | CIFAR-10 | 61.68 | 93.57 | 80.24 | 71.76 |
| | | CIFAR-100 | 64.18 | 92.77 | 80.56 | 72.43 |
| | 2 → 4 | MNIST | 92.74 | 17.24 | 12.27 | 4.27 |
| | | CIFAR-10 | 63.52 | 92.14 | 79.80 | 70.52 |
| | | CIFAR-100 | 66.05 | 91.21 | 79.57 | 70.16 |
| | 8 → 10 | MNIST | 86.77 | 36.61 | 30.79 | 15.10 |
| | | CIFAR-10 | 24.29 | 80.81 | 48.73 | 31.74 |
| | | CIFAR-100 | 38.89 | 59.12 | 64.07 | 43.20 |
| | 9 → 11 | MNIST | 77.19 | 31.40 | 18.79 | 4.32 |
| | | CIFAR-10 | 49.65 | 76.61 | 50.48 | 25.57 |
| | | CIFAR-100 | 35.61 | 68.40 | 55.67 | 31.59 |
| | 2 → 5 | MNIST | 81.11 | 13.09 | 6.74 | 2.24 |
| | | CIFAR-10 | 37.16 | 88.70 | 67.99 | 57.24 |
| | | CIFAR-100 | 39.60 | 86.75 | 70.00 | 58.90 |
| | 7 → 10 | MNIST | 85.04 | 33.28 | 19.26 | 4.59 |
| | | CIFAR-10 | 20.67 | 69.49 | 34.65 | 17.18 |
| | | CIFAR-100 | 30.00 | 48.19 | 53.16 | 26.97 |
| | 1 → 5 | MNIST | 69.44 | 10.36 | 5.38 | 1.56 |
| | | CIFAR-10 | 39.49 | 76.98 | 48.11 | 36.38 |
| | | CIFAR-100 | 36.94 | 72.48 | 51.03 | 38.75 |
| DiNO-S | 2 → 3 | MNIST | 93.04 | 58.24 | 37.95 | 27.62 |
| | | CIFAR-10 | 86.16 | 94.11 | 82.37 | 75.26 |
| | | CIFAR-100 | 86.39 | 93.78 | 82.28 | 75.29 |
| | 3 → 4 | MNIST | 92.33 | 62.78 | 38.18 | 27.52 |
| | | CIFAR-10 | 84.70 | 94.37 | 81.93 | 74.69 |
| | | CIFAR-100 | 83.72 | 94.10 | 82.02 | 74.59 |
| | 4 → 5 | MNIST | 91.64 | 57.39 | 36.97 | 26.02 |
| | | CIFAR-10 | 70.87 | 93.65 | 80.38 | 73.84 |
| | | CIFAR-100 | 71.51 | 92.98 | 79.96 | 73.54 |
| | 9 → 10 | MNIST | 83.39 | 38.85 | 20.20 | 13.10 |
| | | CIFAR-10 | 45.69 | 88.70 | 61.71 | 50.46 |
| | | CIFAR-100 | 60.57 | 76.58 | 76.77 | 61.29 |
| | 1 → 3 | MNIST | 90.60 | 22.30 | 11.76 | 5.47 |
| | | CIFAR-10 | 78.51 | 89.72 | 74.58 | 65.04 |
| | | CIFAR-100 | 79.80 | 89.28 | 74.75 | 64.92 |
| | 3 → 5 | MNIST | 87.54 | 24.55 | 11.93 | 6.67 |
| | | CIFAR-10 | 63.66 | 87.17 | 66.16 | 58.36 |
| | | CIFAR-100 | 64.26 | 84.40 | 66.43 | 58.51 |
| | 2 → 4 | MNIST | 90.54 | 19.14 | 9.99 | 4.99 |
| | | CIFAR-10 | 62.32 | 88.03 | 68.53 | 59.23 |
| | | CIFAR-100 | 64.89 | 86.98 | 68.54 | 59.15 |
| | 8 → 10 | MNIST | 80.88 | 22.27 | 10.30 | 6.25 |
| | | CIFAR-10 | 25.67 | 85.07 | 48.44 | 35.42 |
| | | CIFAR-100 | 29.81 | 67.51 | 67.59 | 47.97 |
| | 9 → 11 | MNIST | 27.79 | 9.93 | 7.30 | 1.67 |
| | | CIFAR-10 | 15.94 | 59.66 | 19.22 | 7.62 |
| | | CIFAR-100 | 15.71 | 40.73 | 32.06 | 12.17 |
| | 2 → 5 | MNIST | 82.67 | 10.77 | 5.85 | 2.85 |
| | | CIFAR-10 | 49.78 | 73.83 | 46.89 | 38.80 |
| | | CIFAR-100 | 48.24 | 67.62 | 46.85 | 38.36 |
| | 7 → 10 | MNIST | 75.50 | 15.89 | 10.43 | 4.24 |
| | | CIFAR-10 | 17.75 | 76.55 | 36.68 | 21.94 |
| | | CIFAR-100 | 19.13 | 53.86 | 55.80 | 33.79 |
| | 1 → 5 | MNIST | 68.07 | 11.29 | 6.29 | 1.74 |
| | | CIFAR-10 | 49.25 | 56.93 | 31.06 | 22.86 |
| | | CIFAR-100 | 47.81 | 47.83 | 30.78 | 21.78 |

