# OpenReview forum: "Detecting and Approximating Redundant Computational Blocks in Neural Networks"
_ICLR.cc/2025/Conference — Submitted to ICLR 2025_

### Official Review · Reviewer_2UjY · 2024-10-27

**Soundness:** 2
**Presentation:** 3
**Contribution:** 2
**Rating:** 3
**Confidence:** 4

**Summary:**

This paper introduces a framework to optimize deep neural networks by detecting and approximating redundant computational blocks, aiming to reduce computational complexity without sacrificing performance. The authors propose a metric called Block Redundancy (BR), which identifies components that do not contribute significantly to the network’s final output.  Using this metric, they develop Redundant Blocks Approximation (RBA), a method that approximates these redundant blocks through simple transformations.

**Strengths:**

- The paper is clearly written and easy to follow.
- The idea of BR and RBA approach is simple and intuitive

**Weaknesses:**

- The paper aims to develop an algorithm for training-free compression of visual transformers. However, the related work appears to be lacking. Several methods such as [1, 2] that achieve compression with minimal training are missing.

- Comparison with state-of-the-art methods is missing. Benchmarking against established compression techniques is essential for positioning RBA’s efficacy relative to existing approaches.

- The experimental evaluation is limited to small datasets, making it difficult to assess RBA’s robustness for larger, more complex datasets such as ImageNet. Demonstrating generalization on larger datasets would strengthen the claim of RBA’s scalability and efficiency.

References

[1] Zhang, Hanxiao, Yifan Zhou, and Guo-Hua Wang. "Dense Vision Transformer Compression with Few Samples." Proceedings of the IEEE/CVF Conference on Computer Vision and Pattern Recognition. 2024.

[2] Bai, Shipeng, et al. "Unified data-free compression: Pruning and quantization without fine-tuning." Proceedings of the IEEE/CVF International Conference on Computer Vision. 2023.

**Questions:**

Please refer to Weaknesses

---

> ### Author Response · Authors · 2024-11-24
>
> We thank the reviewer for their insightful comments and constructive suggestions. In response, we have updated the manuscript (Table 1) to include the requested clarifications and added results using a larger dataset (ImageNet-1K), demonstrating the method's applicability to large-scale datasets. In Table 3, we also conducted new experiments to showcase the method's generalization across different datasets within the same architecture. Below, we address each point raised:
>
> * **W1**: We appreciate the references provided by the reviewer, and we updated the related section with missing citations.
> * **W2**: We understand the reviewer's observation regarding the lack of comparison with other techniques. However, our proposed method is entirely training-free, fine-tuning-free, and architecture-agnostic. In contrast, existing methods in the literature rely on at least some degree of fine-tuning or additional training or are tailored to specific architectures (as [1,2]). Consequently, a direct comparison may not provide a fair or meaningful evaluation. That said, we would be glad to include comparisons with analogous approaches if they exist. If the reviewer is aware of specific works that align with our method's constraints, we would greatly appreciate their suggestions for incorporating them into our analysis.
> * **W3**: We have updated Table 1 to include results on the ImageNet-1K dataset. By incorporating these results, we hope to strengthen the experimental evaluation and address the reviewer's concern regarding the generalization of RBA to large-scale datasets. The updated results demonstrate the effectiveness of our method in achieving significant computational savings while maintaining strong performance, further validating its applicability to complex, real-world scenarios.

---

> > ### Comment · Reviewer_2UjY · 2024-11-30
> > **Official Comment by Reviewer 2UjY**
> >
> > I thank the authors for the detailed revisions and for including the ImageNet results. However, evaluating the effectiveness of the proposed method remains challenging without proper comparisons to relevant baselines. There are several works, such as [1,2], that achieve network pruning either without fine-tuning or by fine-tuning on a very small subset of data. It would strengthen the paper significantly if the authors compared their method to such approaches. Additionally, understanding the trade-offs between accuracy, compression, and fine-tuning time by RBA and other relevant SOTA methods would also be valuable.
> >
> > [1] DFPC: Data flow driven pruning of coupled channels without data. Narshana et al. 2023 ICLR
> >
> > [2] Spdy: Accurate pruning with speedup guarantees. Frantar et al. 2022 ICML

---

### Official Review · Reviewer_tLds · 2024-10-28

**Soundness:** 2
**Presentation:** 3
**Contribution:** 2
**Rating:** 3
**Confidence:** 4

**Summary:**

The paper is interested in finding redundancy within deep neural network models and approximating redundant blocks using a linear transformation. The paper first introduces the Block Redundancy (BR) metric, that computes the MSE between representations of two consecutive blocks on a subset of training data, to evaluate changes in the internal representation of the models. Then, it proposes to approximate highly redundant blocks by finding a good linear transformation between the redundant representations. The linear transformation is found by solving a least square problem between the inputs and outputs of the redundant blocks. Finally, experiments show that using the approximated transformation on redundant blocks preserves performance while reducing number of parameters.

**Strengths:**

- **Significance**: The motivation and research questions of the paper can lead practical impacts. Finding redundancies and approximations for large pretrained models can reduce the number of parameters and make them lighter and more efficient at inference. To this end, the paper shows that we can find such simple approximations while preserving downstream performances.
- **Quality**: The paper presents experiments on multiple pretrained Transformer models and multiple image datasets.
- **Clarity**: The organization of the paper and the writing are clear. I appreciate the "Takeaway" part after each experimental subsection.

**Weaknesses:**

**Originality**:
  - The first part of the paper, regarding the similarities between inner representations of pretrained foundation models, echoes previous studies on the same topic such as Nguyen et al. (2020), as mentioned in the paper. The main difference in this first part, is that the study is done on larger transformer-based foundation models as opposed to convolutional architectures, but the same insights are found, so this is a small contribution in my opinion.
  - It is not clear to what aim the paper introduces this "BR" metric. Why not consider other established similarity metrics such as CKA (Kornblith et al., 2019), or directly the cosine similarity of consecutive representations, for instance ? Why should we use the BR metric instead of other metrics ? In other words, what is measured by BR that is not measured by other metrics ? Furthermore, the similarities between blocks are shown with pairwise cosine similarities in Figure 2 and Section 4.1. Why not use the BR metric in that case ?

**Quality**:
  - There are two weaknesses with the BR metric that are not discussed in the paper: i) it can only be computed between representations of same dimensions, this is usually the case in transformer-based architectures, but not for different kind of architectures, such as CNNs for instance ; ii) the metric is sensitive to different scaling for the representations. That means, for instance, that if the transformation between the two representations is only a linear rescaling by a matrix $A$, such as $h^{(b)}(x) = A h^{(b-1)}(x)$, then $BR(b) = -\frac{||A - I||\_2}{|D\_{sub}|} \sum\_{x \in D\_{sub}} ||h^{(b-1)}(x)||\_2$ which can be very low, even though the overall transformation can be easily approximated linearly. So I'm not sure the BR metric is a good proxy metric to evaluate if the blocks can be linearly approximated or not.
  - The link between the BR metric and the RBA approach is actually not that clear in the paper. Do we observe that redundant blocks found by BR are actually the ones we can "remove" by RBA ? In experiments in Table 1, results when applying RBA to a lot of different blocks are shown, but I don't see a link with actual values of BR for these blocks.

**Significance**:
  - The experiments are only conducted on small scale datasets, the study would be more meaningful and significant with bigger datasets like ImageNet, at least, to evaluate scalability.
  - While the idea of approximating whole blocks to reduce number of parameters is conceptually interesting, is it better than pruning ? Do we achieve a better reduction of number of parameters while preserving performance ?
  - If I understand correctly, the RBA are computed in closed form using 3000 training samples of the *corresponding* dataset the model is then evaluated on. To what extent these approximations transfer from one dataset to another ? If we use 3000 images from cifar10 to approximate some layers, do we preserve performance on ImageNet ?

**Questions:**

I've written specific questions in the Weaknesses part, please refer to that. I've tried to compile some of the questions below:
- Why not consider the BR metric in the pairwise similarity matrices shown in Figure 2 ?
- Is there a link between BR and the preservation of performance after RBA in Table 1 ?
- If we compute RBA with data from one dataset, does it transfer to a different dataset ?
- Are the values shown for "Params" in Table 1 including the RBA parameters of the approximated block(s) ?

---

> ### Author Response · Authors · 2024-11-24
> **Response 1/2**
>
> We thank the reviewer for their constructive feedback. In response, we have updated the manuscript with the requested clarifications and added results using a larger dataset (ImageNet1k), demonstrating the method's applicability to larger datasets. Additionally, we conducted new experiments to demonstrate the method's generalization across different datasets using the same architecture. Finally, we would like to clarify that the proposed method is not restricted to approximating two consecutive blocks; it can also approximate multiple, non-consecutive blocks. The number of approximations is limited only by the total number of available blocks. For instance, one could approximate $b_i$ to $b_{i+3}$ with $i=1$, and later $b_i$ to $b_{i+1}$ with $i=8$. Below, we address each identified weakness and question in detail:
>
> **Originality**:
>
> 1. We appreciate the reviewer’s observation regarding the similarity between the two approaches studied. However, we would like to emphasize that our work extends the findings of Nguyen et al. (2020) to pre-trained foundation models, such as transformers, rather than being limited to convolutional architectures trained from scratch. Furthermore, the first part of the paper serves to contextualize the motivation behind the proposed framework, demonstrating how similarities across layers can be effectively leveraged to enhance the efficiency of large-scale models. This contextualization is integral to framing the broader applicability and impact of our approach.
> 2. We evaluated the representations using alternative metrics, finding that the results remain consistent across them. However, we note that: *(a)* **CKA** is computationally more expensive, especially when applied systematically across all blocks of a large model. *(b)* **Cosine similarity** measures the alignment of vectors but does not account for the magnitude of changes in representations. In contrast, BR uses the negative Mean Squared Error (MSE), which considers both the direction and magnitude of changes in representations. This aspect is particularly important in our context, as we aim to quantify the extent of transformation within the model. To summarize, the BR was chosen for its simplicity and computational efficiency. We have updated **Figure 2** to display the BR matrix instead of the cosine matri,x and the cosine matrices have been moved to the appendix for completeness.
>
> **Quality**:
>
> - We acknowledge the reviewer’s concern regarding the limitations of the BR metric. In the additional experiments we are running, that adopt models with different layer sizes, we addressed this by utilizing the SVCCA metric [1], which provides a complementary perspective. This can be easily done since the proposed framework allows the integration of alternative metrics and transformations based on specific task requirements.
> - To improve clarity, we have included Figure 6 in the manuscript, which visually illustrates the connection between the BR metric and the RBA approach. The results presented in Figure 6 were obtained using the ImageNet-1K dataset with the ViT small model. We believe this addition helps to better contextualize the relationship and enhances the overall understanding of the methodology.

---

> > ### Author Response · Authors · 2024-11-24
> > **Response 2/2**
> >
> > **Significance**:
> >
> > - Regarding the significance of our work, we have expanded our evaluation to include results on the ImageNet-1K dataset, which are now presented in Table 1. Additionally, we have included Figure 6 to further illustrate and support our findings. We believe these additions provide a more comprehensive demonstration of our approach's applicability and impact.
> > - We would like to clarify that our method offers notable advantages over pruning techniques, particularly in terms of trainable parameters. Specifically, the RBA approach eliminates the need for additional training or fine-tuning, as it relies solely on the estimation of a transformation (e.g., a linear transformation) between two spaces. This simplicity is a key strength of our method, as highlighted by the reduction in trainable parameters shown in Table 1. In contrast, existing pruning methods typically require some level of fine-tuning or training, even when minimal, which can add complexity and computational overhead.
> > - We agree with the reviewer that evaluating the transferability of the estimated transformation across different datasets using the same architecture is both an interesting and valuable question. To address this, we conducted additional experiments, which are presented in Table 3. The results show that fitting the transformation on one dataset and applying it to a different dataset for downstream tasks yields very good results. However, when considering simpler datasets such as MNIST, the transformation does not generalize as effectively. These results are consistent across different architectures, further supporting the similarity analysis presented in Section 4.1. This highlights the robustness and adaptability of the proposed method. The complete results for all other datasets and architectures will be eventually included in the camera-ready version.
> >
> > **Questions**:
> >
> > - **Q1 (BR in figure 2)**: In response to the reviewer’s comment on the BR metric in Figure 2, we initially adopted the cosine metric for this plot because, as noted, the results remain consistent across different metrics. However, the cosine plots were chosen originally due to their more interpretable score ranges. To enhance clarity and better align with the main focus of the paper, we have updated Figure 2 to display the BR matrix instead of the cosine matrix. The original cosine-based results have been moved to the appendix for reference.
> > - **Q2 (BR and RBA link)**: The connection between the BR metric and the RBA approach is now clearly illustrated in Figure 6. This figure demonstrates that approximating similar representations yields improved results, highlighting the importance of redundancy in the process. Moreover, it shows that approximating highly redundant blocks leads to better performance compared to approximating less redundant blocks.
> > - **Q3 (Transferability)**: We present the new results in Table 3, where we show the ability of RBA to transfer effectively across datasets within the same architecture. Our experiments demonstrate that the estimated transformations retain their effectiveness when applied to similar datasets, while performance may vary for datasets with significantly different characteristics (e.g. MNIST and CIFAR100).
> > - **Q4 (Params column)**: The "Params" values reported in Table 1 already include the RBA parameters associated with the approximated blocks. This ensures that the parameter count accurately reflects the full scope of the model, including the additional parameters introduced by the RBA approach.
> >
> > We would like to thank the reviewer again for their valuable feedback and constructive suggestions. The time and effort invested in reviewing this work are greatly appreciated and we hope our responses have adequately addressed your concerns. We welcome any further feedback or suggestions to strengthen the work further.
> >
> > ---
> >
> > [1] Raghu, Maithra, et al. "Svcca: Singular vector canonical correlation analysis for deep learning dynamics and interpretability." Advances in neural information processing systems 30 (2017).

---

> > > ### Comment · Reviewer_tLds · 2024-11-27
> > >
> > > I thank the authors for their detailed answer and taking the time to update their paper with additional experiments. I appreciate the results on ImageNet and the transferability experiments, although I would have preferred a CIFAR -> Imagenet experiment (as I mentioned) instead of CIFAR10 -> CIFAR100.
> > >
> > > However, I still have concerns that I expressed in my review for which I was either not convinced or were not addressed:
> > > - The method has the inherent weakness of being only applicable to transformer architectures.
> > > - The question of sensitivity to rescaling is a problem. I'm not convinced that MSE is the best metric to estimate if a layer can be approximated by a linear representation because of that.
> > > - I'm not convinced by the discussion regarding comparison with pruning methods.
> > >
> > >    First, I do not agree that all pruning methods require fine-tuning or training. From a quick search, Singh \& Alistarh (2020) [A] presents results in "one-shot pruning" without re-training the pruned network. Therefore, it is a setting considered in pruning, which could serve as a ground for comparison with the proposed method.
> > >
> > >    Second, while I agree that the method has the advantages of computing linear approximations in closed form without retraining the model, and that comparisons should be fair, it is currently very difficult to evaluate the effectiveness and the significance of the "reduction in trainable parameters shown in Table 1" in a vacuum, without a comparison.
> > >
> > > Furthermore, I also agree with the weakness raised by reviewer tZQp about the method requiring a NAS. There is currently no obvious way to find the good set of layers to approximate, thus requiring to compute a lot (maybe all) of possible combinations. This can be seen in Figure 6, with ViT-S on ImageNet. Approximating 4-> 5 gives lower performance thant 2->3, even though BR is better for 4->5.
> > >
> > > [A] Singh & Alistarh "Woodfisher: Efficient second-order approximation for neural network compression". NeurIPS 2020 (https://arxiv.org/pdf/2004.14340)

---

### Official Review · Reviewer_tZQp · 2024-10-29

**Soundness:** 2
**Presentation:** 2
**Contribution:** 3
**Rating:** 6
**Confidence:** 4

**Summary:**

Based on the observation that multiple blocks in neural networks produce similar representations, the paper identifies these similar blocks (basically a vit encoder layer) and approximate the later blocks from previous blocks with similar representations. The paper uses the inverse of MSE between [CLS] tokens of transformers to identify redundant blocks. The redundant blocks are estimated using a linear transformation. The experiments showed that there is a increase in accuracy when approximating the redundant blocks in some cases but, in some cases accuracy decreases as well.

**Strengths:**

The paper studied an important problem of removing redundant blocks from a vision transformer. The study approximated all intermediate redundant blocks which can remove multiple maybe less redundant layers that still ultimately leads to a similar representation as the preceding layers. The paper perform experiments to measure accuracy on removing different redundant blocks. The writing is written in simple grammar making it easy to understand. The observation that redundant blocks are model specific and not data specific is important as it allows building an architecture that is inherently faster than the base model with similar capabilities. The paper provides detailed reproducibility statement in its appendix.

**Weaknesses:**

1. Novelty of Block Redundancy: Block redundancy is just the negative MSE of block i and block i+n outputs, cant an MSE just do the same job albeit a lower mse meaning more redundancy?
2. Tables 1,2 and 3 show that the best architecture needs to be searched and there is no singular dataset agnostic architecture that consistently maintain or improve the performance as shown in figures 2 and 6, although the change in accuracy is less, in cases where the drop in accuracy is important, the complexity of finding the optimal architecture increases.  This could be because of two reasons, a.)The BR is too simple that it isnt a suitable metric to rank redundant blocks. b.) The linear approximation performed might not be optimal.

3. This paper is not the first to try replacing redundant parts of a vision transformer. Venkataramanan et al [1] does linear approximation of redundant attention layers. While both papers are different as replacement learning focuses on replacing complete encoder blocks, the linear approximation and redundancy metric [1] can easily be extended to block level replacement. How does the BR and linear approximation compare to Venkataraman et al's approach when performing block approximation? basically we need to compare the accuracy and throughput (if significant).

4. No results on downstream tasks like segmentation and object detection. This is important to understand the effectiveness of replacement of redundant blocks for tasks other than classification. The paper could apply the technique on Uformer on ADE20K dataset for Segmentation like in [1] and DETR on Pascal VOC / COCO for object detection. Feel free to use any other models or benchmarks other than specified.

Nitpick: In figure 6 in the row of DeiT-S there is are hidden column names behind the image, that could be cleared in case of preparing camera ready.

[1] Shashanka Venkataramanan, Amir Ghodrati, Yuki M Asano, Fatih Porikli, & Amir Habibian (2024). Skip-Attention: Improving Vision Transformers by Paying Less Attention. In The Twelfth International Conference on Learning Representations.

**Questions:**

I would improve my score if the following questions are answered:
1.  The need for BR and not simply using MSE? I suggest that the paper removes the contribution of BR and just stick with MSE or negative MSE.
2. Some results on segmentation and object detection? Whats important is whether we can get a dataset agnostic architecture of modern transformers with lesser parameters. If time does not permit both segmentation and object detection results performing only segmentation is also acceptable, but, it is important to compare the results with [1].
3. Can you provide ablations on BR and Linear Approximation with [1]? This is important to justify the design choices made in this paper.

---

> ### Author Response · Authors · 2024-11-24
>
> We thank the reviewer for their insightful comments and constructive suggestions. Below, we address each point raised:
>
> 1. **W1 and Q1 (BR metric)**: We appreciate the reviewer’s concern regarding the use of the BR metric nomenclature. We have revised our contribution to focus not on proposing a new metric but on demonstrating that a simple metric is sufficient to capture the similarities. However, we believe that this terminology improves the readability of the manuscript and supports our goal of presenting the work in a clear and accessible manner. Thus, we maintained the acronym. We hope this explanation addresses the concern raised.
> 2. **W2 (consistent results across different settings)**: We agree with the reviewer that the linear approximation used in our approach may not be optimal in all scenarios. However, the primary objective of this paper is to introduce a flexible framework that can be seamlessly integrated into a variety of pipelines. The transformation used in the framework is modular and can be easily replaced with more advanced options, allowing users to adapt it to their specific needs.
> 3. **W3 (comparison with Venkataramanan et al [1])**: We thank the reviewer for pointing out this related work. After a detailed analysis, we have identified several key differences between our approach and the method proposed in [1]. The primary distinction lies in the fact that, as stated in the appendix of [1] (Section "Architecture/Image Classification"), the authors re-train the networks from scratch after computing the SKIPAT. Additionally, [1] introduces a method for skipping MSA computation in one or more layers of a transformer, while our approach provides a general method to approximate any layers (not limited to the attention heads) in any architecture (not limited to transformers). Also, the SKIPAT parametric function in [1] employs two linear layers and a depth-wise convolution (DwC) in between, while our method involves only a linear transformation. We believe that comparing a method that requires training and is architecture-specific with one that does not and is architecture-agnostic is not fair. Nevertheless, we remain open to adding comparisons with methods that do not require further training and are architecture-agnostic if the reviewer can provide those references since, to the best of our abilities, we could not find any.
> 4. **W4 and Q2 (other downstream tasks)**: As mentioned in the limitations section, we agree with the reviewer that including results on other downstream tasks could enhance the significance of the paper. Since our method can be applied to other tasks without any inherent limitations, we are currently conducting additional experiments using AutoEncoders for image reconstruction and ResNets, which will be eventually included in the camera-ready version.
> 5. **Q3 (ablations)**: We kindly ask the reviewer to clarify Question 3, as we did not fully understand the specific ablation being requested. However, we are happy to provide any additional results as needed.
>
> We thank again the reviewer for their thoughtful and constructive feedback. We hope our responses have adequately addressed your concerns, and we welcome any further feedback or suggestions to strengthen the work further.

---

> ### Comment · Reviewer_tZQp · 2024-11-24
> **Clarifications regarding my review**
>
> Thank You for your detailed response to my review, while I am happy with the response for questions 1 and 4. I need to clarify my points 2,3 and 5 for completing the novel research conducted.
>
> Clarification:
>       The most significant drawback I noticed in this approach is stated in my weakness point 2, the results are not consistently maintaining or improving the performance. While I dont expect the design choices (BR and RBA) to reflect ideal results, to ensure the proposed method is best choice currently possible, I expect the paper to perform ablations comparing BR and RBA with other possible approaches. For BR, as reviewer  tLds has suggested, I recommend comparing with other  distance metrics like cosine similarity. For RBA I understand the core motive of the paper is to state "even a simple linear transformation is enough to approximate redundant blocks" the proof shown to back that statement are tables 1,2 and 3 primarily. While this proof is partially acceptable, it is important to estimate the loss in performance by comparing with say, nonlinear approximation for instance. For simplicity, I asked the authors to compare the "approximation" technique used by venkatraman et al. approach. To again clarify, since I want the comparison to be done only with BR and RBA i still find venkatraman et al's approach eligible to be compared. Moreover, I dont support the idea that the proposed approach is optimization free, RBA requires the latent space to be estimated, is the estimation performed on random data? if not, I consider this method to involve training. Moreover, if its random, I am curious to know the reason why that "design choice" was made.
>
> Point by Point clarification:
>
> 2. While I appreciate the proposal made by the paper, I don't agree with the point "can be easily replaced with more advanced options, allowing users to adapt it to their specific needs." The paper must present why they think RBA and BR is suitable as an initial approach for approximating redundant blocks, not leave that to the users.
> 3. I hope I have made a detailed suggestion for this point in the above clarification, I am available to discuss this further for the betterment of the work.
>
> 5.) The design choices are BR,RBA primarily, I want some ablations to compare these proposals with other simple replacements, as the authors themselves accept the flexibility of the approach to switch BR and RBA approximations with more advanced choices.
>
> I am still open to updating my score, but the rebuttal doesnt satisfy me enough to raise my score to a 6 or 8, therefore I await more actions to improvise and strengthen the proposed approach.
>
> General Suggestion to paper authors: It would be simple for reviewers to view the changes made to the paper if they are highlighted red. This highlights can be removed during camera ready.

---

> > ### Author Response · Authors · 2024-11-26
> >
> > We sincerely thank the reviewer once again for their valuable suggestions to enhance our work and for their effort in providing detailed and prompt feedback. Below, we address each point in detail:
> >
> > 1. **Comparison with Other Metrics:** As suggested by the reviewer and reviewer tLds, we have provided an ablation study on the ImageNet-1k dataset, presented in Figure 7. We compare the BR metric with cosine similarity, CKA, and SVCCA. The results indicate that using the BR or cosine similarity enhances the visualization of block structure, making it easier to identify highly redundant blocks. However, the findings highlighted in orange demonstrate that the BR metric more effectively captures redundancies and non-redundancies. For instance, it is the only metric that indicates approximating blocks 7 → 10 is suboptimal, as it leads to lower accuracy. Additionally, when considering the same number of blocks, the BR metric shows that approximating blocks 2 → 5 yields more favorable results. Thus, we believe that the BR metric is the most suitable one.
> > 2. **Other Approximation Techniques:** We conducted an ablation study comparing various approximation techniques to further support our claim that using BR and RBA is the most effective solution and that a simple transformation suffices to approximate redundant blocks. Table 4 presents results using RBA, a simple MLP, and a more complex MLP (referred to as Res-MLP). The MLP is a lightweight multi-layer perceptron, while the Res-MLP consists of a feedforward network with normalization and a residual connection (Please refer to the appendix for complete details). We did not include the SKIPAT module as a translator since it involves a 2D convolutional module, which is not applicable since we are estimating transformations on 1D tensors. The experiments were conducted on ImageNet1k using the ViT-small model and three different seed values. It is also important to emphasize that RBA is significantly faster than the other methods since it requires no backpropagation. In contrast, the MLP and Res-MLP are trained for 300 steps using a learning rate of 1e-3 and the Adam optimizer. Results show that employing a simple linear transformation to approximate redundant layers is the optimal choice in most scenarios. However, as expected, the more blocks are approximated, the less linearly correlated they become, making more complex approximations more effective (see 1 $\rightarrow$ 5 in Table 4).
> > 3. **Optimization-Free Methods:** We appreciate the reviewer's feedback regarding our claim of a "training and fine-tuning free" approach. To clarify, while the estimation process relies on a random small subset of the training data (not random data), it is performed via a linear transformation that is computed in closed form. This process does not involve gradient computation or backpropagation. Our assertion of being "training-free" is based on the absence of these optimization-based procedures.
> >
> > We again sincerely thank the reviewer for their feedback. As suggested, all changes made during the rebuttal process have been highlighted for clarity. We are grateful for the reviewer's thorough effort and hope that our clarifications have adequately addressed their concerns and fulfilled their requests. Nevertheless, we remain open to providing additional comments or experiments if needed.

---

> > > ### Comment · Reviewer_tZQp · 2024-11-27
> > > **Updation of Scores**
> > >
> > > The paper adressesses the most pressing issues from my comments. I find the newer version to represent the claim "linear approximation is enough for redundant block removal", I also accept point 3 to some extent. Given these new updations to the paper, I update my score to 6. Reason for not giving a higher score: In a practical scenario the optimal network still needs a NAS.
> > >
> > > Additional suggestions to the paper authors: As the authors have previously obliged to provide experiments on autoencoders and resnets by the end of the camera ready, if the paper is unable to produce those results, I suggest the authors to tone down the claim of broad applicability. More like "will" to "may".

---

> ### Comment · Reviewer_tZQp · 2024-11-27
> **Rethinking my assessment**
>
> While I still standby my argument that the approach requires a NAS, I am also interested in knowing the time taken to each removal (ex. 1 ->5), I don't want any new experiments to be added, but I may reassess my review if the time required for NAS is tolerable. Also, how would a NAS look like in this case, for example we have 1 -> 5 , 6->8 as two set of similar blocks, from your experience of conducting experiments presented in tables 1 to 3, will we need to search the subset combinations (i.e 1 -> 5 = {1->5, 2->4,3->5, etc}) or searching a combination between the two set of similar block (1->5 or 6->8) enough.

---

### Official Review · Reviewer_bYFY · 2024-11-01

**Soundness:** 2
**Presentation:** 3
**Contribution:** 3
**Rating:** 5
**Confidence:** 4

**Summary:**

This paper investigates the similarities across different layers in diverse neural architectures and assesses the redundancy of blocks in vision transformer based on the similarities of outputs between two blocks. Ultimately, redundant blocks are replaced with simple linear layers to achieve lightweight models. Experiments on multiple datasets and models demonstrate that the proposed method appears to be effective.

**Strengths:**

1. Code is released.
2. The paper is well-presented and easy to follow.
3. The experimental results and ablation study show the effectiveness.

**Weaknesses:**

1. In Equation 1, when $b=1$, what does $\bf{h} ^ {(0)} (x)$ represent?
2. In Line 147 on Page 3, the authors state that a higher BR indicates a potential redundancy in block $b$. Why is block $b-1$ not considered to be a redundant block? Similarly, in Line 158, why not skip $b _ i$ while retaining any layer or several layers from $b _ {i+1}$ to $b _ {i+n}$?
3. In Table 2, is the retraining step conducted under the 'skip' mode? If not, the accuracy after retraining should be reported.
4. This paper lacks the comparison with other model compression methods, such as pruning techniques.
5. The results on more complex datasets, such as ImageNet, should be reported. Also, can the proposed method be used to other tasks, such as object detection?

**Questions:**

Please see the weaknesses.

---

> ### Author Response · Authors · 2024-11-24
>
> We thank the reviewer for their insightful comments and constructive suggestions. Below, we provide detailed responses to address each of the raised points and clarify aspects of our approach. Additionally, we updated the PDF submission with the corresponding corrections.
> * **Clarification on Equation 1**: Our method focuses on approximating the latent space using internal representations. Specifically, in our setup, we start from block $b=1$. For models such as ViT, which have 12 internal blocks, the indexing runs from 0 to 11. As a result, block 0 is not subject to approximation in our approach; the approximation process begins from block 1. Therefore, when $b=1, h^{(0)}(x)$ refers to the output of the initial (zero-th) block of the transformer model. We hope this clarifies our notation.
> * **Addressing Block Redundancy**: In this paper, we define redundancy in the context of a sequence of blocks rather than individual blocks. Specifically, a sequence of blocks $b_i,…,b_{i+n​}$ is considered redundant if the representation produced by the block $b_{i-1}$ is similar to that produced by the block $b_{i+n}$​. This indicates that the sequence as a whole contributes to redundancy, as the collective sequence leads to a similar final representation. Importantly, redundancy is a property of the sequence as an entirety, not of any single block in isolation. A single block $b_i$​ cannot be definitively labeled as redundant merely because it is part of a redundant sequence. For instance, to better illustrate the reasoning, consider a block followed by its inverse. Their composition is redundant since it is the identity, however the individual blocks are not.
> * **Clarification on ‘Skip' Mode in Table 2**: We apologize for any ambiguity regarding retraining. In Table 2, we present the accuracy results for two distinct inference modes: (1) the “skip” mode, a naive baseline, where entire blocks are bypassed without any approximations, and (2) the “approximate” mode, where blocks are not fully skipped but are instead approximated (i.e., the approach we are proposing). These results are obtained without any additional retraining; the accuracies reflect the performance of the model under these two inference strategies as-is. To avoid any confusion, we have further clarified this point in the text.
> * **Comparison with Model Pruning Methods**: We understand the reviewer’s observation regarding the lack of comparison with pruning techniques. However, our proposed method is training-free, fine-tuning-free, and architecture-agnostic, whereas most methods in the literature rely on at least some degree of fine-tuning, additional training, or tailored to specific architectures. Consequently, a direct comparison may not provide a fair or meaningful evaluation. That said, we would be glad to include comparisons with analogous approaches if they exist. If the reviewer is aware of specific works that align with our method’s constraints, we would greatly appreciate their suggestions to incorporate into our analysis.
> * **Experiments on Complex Datasets and Other Tasks**: We appreciate the reviewer’s valuable suggestion to include additional experiments. To address this, we have extended our evaluation to assess the robustness and adaptability of our approach in more challenging settings. Specifically, we have updated Table 1 in the manuscript to present results on the ImageNet-1K dataset, which provides a more comprehensive benchmark for large-scale image classification tasks. Additionally, we also add in Figure 6, the correlation between results on ImageNet-1K and the BR metric. Regarding other tasks, we agree that results on other downstream tasks could improve the significance of the paper, as we expressed in the limitation section. Since there are no limitations in applying the method to other tasks, we are currently performing additional experiments using AEs for image reconstruction and ResNet-based architectures that will be eventually added to the camera-ready version.
>
>
> ---
>
>
> We sincerely thank the reviewer for their thoughtful and constructive feedback, which has been instrumental in refining our approach and enhancing the clarity and quality of our manuscript. We hope our detailed responses have adequately addressed your concerns, and we welcome any further feedback or suggestions to strengthen the work further.

---

> ### Comment · Reviewer_bYFY · 2024-11-30
>
> I appreciate the authors for their thoughtful and detailed responses. The clarifications for Q1, Q2, and Q3 have addressed my concerns. Additionally, the important experiments on ImageNet (Q5) has been added to the revised manuscript (in Table 1).
>
> However, the comparison with other methods is still missing, which undermines the persuasiveness of the proposed method. This issue is also raised by reviewer tLds.
>
> In conclusion, I will keep my original rating.

---

### Author Response · Authors · 2024-11-24
**General response**

We would like to express our sincere gratitude to the reviewers for their thoughtful feedback and constructive comments. We deeply appreciate that all reviewers found the paper to be well-organized and easy to follow. We are particularly encouraged by the recognition that “the experimental results and ablation study show the effectiveness” of our approach (bYFY). Additionally, we thank the reviewers for acknowledging the practical implications of the motivation and research questions addressed in our work (tLds). Lastly, we are grateful for the appreciation of our “simple and intuitive” idea (2UjY).

However, we would like to take this opportunity to clarify a couple of key points that were raised by different reviewers but can be useful to everyone to better clarify the proposed work:

* **Training-Free and Tuning-Free Methodology**: Our approach is entirely training-free and tuning-free. Unlike existing methods, we do not involve any training or fine-tuning of the approximated network. Instead, we employ a closed-form estimation of a linear transformation to approximate one or multiple blocks. After the approximations, as usual, a single linear layer classifier is trained on top of the pretrained model to perform the desired downstream task.

* **Flexibility Across Arbitrary Blocks**: The method is highly flexible and can be applied not only between two consecutive blocks but also across any number of arbitrary, non-consecutive blocks. For example, it is feasible to approximate representations from block 2 to block 4 and subsequently from block 6 to block 7, as shown in Figure 1 in the paper.

* **Broad Applicability**: Our method does not make any assumptions on specific layers or architectures. Therefore, it can be applied to any computational block and architecture, for any downstream task.


---



Additionally, we address the reviewer's concerns by performing the **following additional experiments**:
* To enhance the evaluation and address concerns about the generalization of RBA to large-scale datasets, we included results on **ImageNet-1K** in Table 1, demonstrating the method's applicability to larger datasets.
* To further support our approach, we have added Figure 6, which illustrates the **correlation between ImageNet-1K results and the BR metric**. The figure shows that approximating blocks with high BR values preserve comparable accuracy while significantly reducing the parameter count and computation time. Moreover, the results show that approximating more high-BR blocks achieves better accuracy compared to approximating fewer low-BR blocks, which are less redundant.
* In Table 3, we present **additional experiments** to evaluate the model's ability to approximate representations based on a transformation derived from a different dataset but applied using the same architecture. Except for MNIST—which may be too simplistic to generalize effectively—our findings consistently **show that a simple linear transformation can be shared across all tokens and datasets**, highlighting the robustness and flexibility of the proposed method.

---

> ### Author Response · Authors · 2024-11-26
> **Changelist**
>
> We sincerely thank all the reviewers for their constructive feedback on our work, suggesting actionable modifications to improve the paper further.
>
> Following their advice, we list here the main changes we adopted in the manuscript:
>
> - Evaluation of the proposed method on ImageNet1k (*Table 1*)
> - Experiment to show that a simple linear transformation is shared across different datasets within the same architecture (*Table 3*)
> - Ablation on the translation complexity, showing that a linear transformation is enough (*Table 4*)
> - Further demonstration that the proposed metric (BR) correctly identifies redundant blocks (*Figure 6 and Figure 7*)
> - Further clarified that our method is training-free, fine-tuning free, and architecture agnostic.
> - Further clarified that our method can be applied between an arbitrary number of blocks.
> - Clarified the BR contribution.
> - Updated the related work section adding [1,2,3].
> - Updated Figure 2 with the BR metric and moved the cosine similarity block-by-block similarities to the appendix.
>
>
> ---
>
> [1] Shashanka Venkataramanan, Amir Ghodrati, Yuki M Asano, Fatih Porikli, & Amir Habibian (2024). Skip-Attention: Improving Vision Transformers by Paying Less Attention. In The Twelfth International Conference on Learning Representations.
>
> [2] Zhang, Hanxiao, Yifan Zhou, and Guo-Hua Wang. "Dense Vision Transformer Compression with Few Samples." Proceedings of the IEEE/CVF Conference on Computer Vision and Pattern Recognition. 2024.
>
> [3] Bai, Shipeng, et al. "Unified data-free compression: Pruning and quantization without fine-tuning." Proceedings of the IEEE/CVF International Conference on Computer Vision. 2023.

---

### Meta-Review · Area_Chair_kgAs · 2024-12-23

**Metareview:**

The authors propose a method to identify redundant blocks in neural networks, and replace them with a simpler, less costly linear transforms. The Block Redundancy (BR) metric measures the -MSE between output representations of blocks, with a higher value indicating minimal change in outputs. Redundant blocks can be replaced by a linear transform on the outputs of the preceding similar block, thereby increasing the model efficiency at minimal drop in accuracy.

The final reviewer ratings were 3, 3, 5, 6.

The ACs would like to highlight a related area of work that has not been discussed in the submission or the reviews. Note that the models considered in this work are all residual networks (ViTs have skip layers, just like the original ResNets based on convolutions). There is a rich history of prior work showing that ResNets are robust to dropping of entire blocks and that features in ResNets typically change slowly with depth:

1) Residual Networks Behave Like Ensembles of Relatively Shallow Networks (NeurIPS 2016): Shows the entire layers/blocks can be removed in ResNets, without any noticable change to accuracy. Deleting multiple blocks at once increases error smoothly.

2) Convolutional Networks with Adaptive Inference Graphs (ECCV 2018), BlockDrop: Dynamic Inference Paths in Residual Networks (CVPR 2018): Subsets of layers can be completely skipped on a per-input basis.

3) Neural Ordinary Differential Equations (NeurIPS 2018): Explores the parallels between ResNets and ODEs that continuously and smoothly transform the hidden state.

4) Do Residual Neural Networks discretize Neural Ordinary Differential Equations? (NeurIPS 2022): Explores ResNets as discretization of Neural ODEs.

There is a clear parallel between what this submission tries to do and what the above mentioned works show - that layers/blocks in residual networks change features smoothly, have redundancies, and can even be skipped. So the idea of replacing entire layers/blocks with linear transformations should not come as a big surprise.

Futher, the last 2 papers mentioned above could even provide a theoretical backing for the proposed work. This submission is incomplete without an examination of the relationship between such prior work.

The ACs have taken into account the confidential feedback submitted by the authors.

Based on all the reviews and discussions, and the points made above, including the omission of very related work, the ACs do not believe this work in its current form is ready for acceptance.

**Additional Comments On Reviewer Discussion:**

During the review, multiple reviewers asked for results on ImageNet, which the authors provided, and was appreciated by the reviewers. While multiple reviewers asked for comparison to pruning-based networks, the ACs did not think this was a major disqualifying factor, since pruning requires more computation as compared to the proposed approach, and can probably be used in conjunction with the proposed method.

---

### Decision · Program_Chairs · 2025-01-22

Reject